# OptiMind: Teaching LLMs to Think Like Optimization Experts

## Abstract

Mathematical programming – the task of expressing operations and decision-making problems in precise mathematical language – is fundamental across domains, yet remains a skill-intensive process requiring operations research expertise. Recent advances in large language models for complex reasoning have spurred interest in automating this task, translating natural language into executable optimization models. Current approaches, however, achieve limited accuracy, hindered by scarce and noisy training data without leveraging domain knowledge. In this work, we systematically integrate optimization expertise to improve formulation accuracy for mixed-integer linear programming, a key family of mathematical programs. Our OptiMind framework leverages semi-automated, class-based error analysis to guide both training and inference, explicitly preventing common mistakes within each optimization class. Our resulting fine-tuned LLM significantly improves formulation accuracy by 21.4% across multiple optimization benchmarks, with consistent gains under test-time scaling methods such as self-consistency and multi-turn feedback, enabling further progress toward robust LLM-assisted optimization formulation.

## 1 Introduction

Mathematical optimization plays a critical role across many business sectors, from supply-chain management to energy systems to logistics planning, where effective decision-making relies on solving highly complex optimization problems. While practitioners can usually describe these problems in natural language, translating them into precise mathematical formulations that optimization solvers can process remains a skill-intensive bottleneck. Crafting a correct formulation requires precise definition of decision variables, objectives, and constraints, a skill that typically takes years of specialized training in operations research to develop.

Researchers and practitioners have begun exploring whether large language models (LLMs) can automate the formulation task: translating natural language problem descriptions into executable optimization models (Ramamonjison et al., 2022; Tang et al., 2024; Yang et al., 2025b). Success on this front would lower a major barrier to broader use of optimization, democratizing its benefits. Yet current systems remain far from this goal. Accuracy is limited by training data quality, including lack of diversity and frequent errors from synthetic data generation. On the inference side, most existing approaches rely on a simple prompt that includes the natural language problem description and the modeling goal (e.g., generate an integer programming formulation in Pyomo, OR-Tools, or GurobiPy). This leaves out more structured prompting strategies that may yield stronger outcomes. The problem is further compounded by noisy benchmarks, such as ambiguous questions, missing data, and incorrect ground-truth values, where insufficient manual cleaning produces high error rates that obscure true performance. Critically, existing approaches underutilize domain expertise in benchmark quality control, training, and inference.

In this work, we study how *optimization expertise* can be systematically integrated – both at training time and inference time – to improve formulation accuracy at scale. A core component of achieving strong training outcomes is high-quality data. Motivated by our observation that existing LLMs often repeat similar mistakes within each optimization problem class, we first map problems into a fixed set of canonical classes (e.g., set cover, flow shop scheduling), then manually analyze a small subset of representative examples from each class to extract common formulation errors. We then leverages

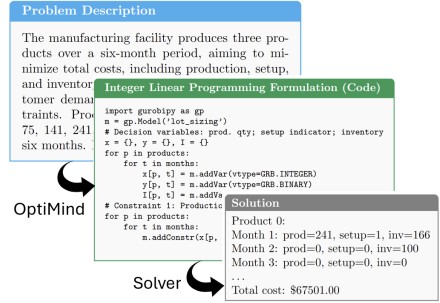

Figure 1: OptiMind high-level overview.

Figure 2: From problem description to solution.

this error analysis to automate the training dataset cleaning pipeline by explicitly prompting strong LLMs to follow class-specific hints and avoid common error patterns, further using self-consistency to improve solution generation quality.

These class-based error analyses not only guide the fine-tuning of strong LLMs, but they can also improve performance at inference: the same formulation hints derived from the training data can be incorporated directly into prompts and help reduce common formulation errors. As LLMs may still generate noisy outputs, producing infeasible models or code with execution errors, we further apply test-time scaling methods such as self-consistency and multi-step refinement when additional compute is available. We henceforth refer to our combined training and inference framework as *OptiMind*, as it leverages domain expertise to improve a critical optimization task.

We evaluate our approach on the recently released GPT-OSS-20B, applying both training-data and inference enhancements. To ensure reliable evaluation, we carefully re-clean three of the most challenging public benchmarks. Across these benchmarks, *OptiMind* improves absolute accuracy over the GPT-OSS-20B base model by between 13% and 21% and consistently outperforms other open-source models of comparable or larger size, while approaching the performance of proprietary frontier models. An ablation study confirms that both training-data and inference components contribute significantly to these improvements. We further show that incorporating hints improves accuracy across multiple models, pointing to the robustness of our approach. Our contributions are summarized as follows:

- We introduce a domain-informed error analysis framework that semi-automatically cleans and improves training data for optimization formulation tasks.

- We train a 20B-parameter GPT-OSS-20B variant on the semi-automatically cleaned data, yielding a strong open-source model for optimization formulation.

- We propose a inference pipeline that integrates class-specific error summaries as error-analysis-based hints, enabling optional test-time scaling with majority voting and self-correction.

- We conduct extensive empirical evaluation on three manually cleaned benchmarks, showing that our *OptiMind* framework substantially improves formulation accuracy, surpasses all open-source baselines of similar or larger size, and achieves performance competitive with much larger proprietary frontier models.

Together, these results demonstrate the importance of domain knowledge in making LLMs more reliable for optimization, advancing the broader goal of intelligent automation in decision-making. We plan to open-source our framework, data, cleaned benchmarks, and error analysis methods to enable further progress in the community.

## 2  DESCRIPTION OF THE FORMULATION TASK

The *formulation task*, which is the focus of this paper, consists in translating natural language problem descriptions into executable optimization models. To make this precise, we first describe the class of optimization models considered.

**Mixed-Integer Linear Programming (MILPs).** MILP is the class of optimization problems where some decision variables are constrained to be integers, and relationships among variables

are linear in the form of constraints and an objective function. A MILP problem is formulated as $\min\{c^\top x : Ax \leq b, x_j \in \mathbb{Z} \ \forall j \in I\}$, where $x \in \mathbb{R}^n$ is the vector of decision variables, $c \in \mathbb{R}^n$ is the cost vector, $A \in \mathbb{R}^{m \times n}$ and $b \in \mathbb{R}^m$ define the linear constraints, and $I \subseteq \{1, \ldots, n\}$ indicates the variables required to be integer-valued. MILPs are very general and used extensively to model complex decision-making problems involving both discrete choices and continuous variables, capturing applications in supply-chain optimization, scheduling, network design, resource allocation, and much more (Fleuren et al., 2013; Kroon et al., 2009; Durán et al., 2007; Lee et al., 2009).

**The Formulation Task.** The input of the task is a complete description in natural language of the decision-making problem that one wishes to model and solve, including all the required data. For example, in the context of manufacturing planning, this data typically includes demand values for the products on different periods, machine capacities or other resource quantities, etc. The desired output of the task is an MILP formulation of the input problem. Concretely, the output format we consider is a Python code that specifies the decision variables, constraints, and objective function of the output MILP; this leverages the strong ability of current LLM models of producing Python code. In addition to defining the MILP formulation, the output code also has commands to execute a solver (e.g., Gurobi (2025)) and a short routine to print the optimal decisions. Running the output code thus produces the complete set of decisions ready to be employed by the user. Fig. 2 illustrates this process, and in Appendix A.3 we provide a complete input/output example.

## 3 RELATED WORK

A growing body of work focuses on building benchmarks for natural language to optimization formulation translation. NL4LP (Ramamonjison et al., 2022) was one of the early benchmarks introduced in a NL4OPT competition, focusing on Linear Programming (LP) problems. Mamo (Huang et al., 2024) expands the scope to MILPs, with Mamo Easy and Mamo Complex reflecting different difficulty levels. NLP4LP (AhmadiTeshnizi et al., 2024) contains LP and ILP problems collected from textbooks and lecture notes. Xiao et al. (2024) released ComplexOR, a benchmark of OR problems collected from both industrial and academic scenarios. IndustryOR (Tang et al., 2024) provides 100 real-world OR problems across eight industries, while OptiBench (Yang et al., 2025b) extends coverage to nonlinear and tabular problems. OptMATH (Lu et al., 2025) further introduces GPT-synthesized benchmark with longer natural language contexts and complex constraints. LogiOR (Yang et al., 2025a) proposes a new optimization modeling benchmark from the logistics domain, containing more complex problems with standardized annotations.

A major limitation in current benchmarks is their high error rate. Both our own experience and a recent survey (Xiao et al., 2025b) reveal frequent issues, including missing data, ambiguous formulations, and incorrect ground-truth answers. We thus have dedicated significant efforts to cleaning some of the benchmarks. In parallel to our efforts, several recent works also focus on improving the quality of existing datasets. SIRL (Chen et al., 2025) releases cleaned versions of NL4OPT, IndustryOR, MAMO-ComplexLP, and MAMO-EasyLP. Yang et al. (2025a) further provide cleaned variants of IndustryOR, ComplexOR, and NL4LP as part of the LogiOR benchmark. Xiao et al. (2025a) examine six mainstream benchmarks, including IndustryOR and MAMO-Complex, and find that about 30-50% of the data are problematic; they discard these instances and release a curated subset containing only validated problems.

Several prompting strategies have been developed to improve formulation accuracy. Chain-of-experts (Xiao et al., 2024) and OptiMUS (AhmadiTeshnizi et al., 2024) adopt agentic frameworks that split the task into specialized LLM agents for modeling, programming, and evaluation. Search-based methods such as AutoFormulation (Astorga et al., 2025) use Monte-Carlo Tree Search to separately construct variables, constraints, and objectives, while pruning equivalent formulations at the symbolic level. More recently, OptiTree (Liu et al., 2025) further decomposes complex optimization tasks into simpler subproblems to improve the overall solution. Multiple works have explored fine-tuning strategies for optimization datasets. ORLM (Tang et al., 2024), ReSocratic (Yang et al., 2025b), and OptMATH (Lu et al., 2025) apply supervised fine-tuning (SFT) on LLM-synthesized datasets to improve performance on IndustryOR, OptiBench, and OptMATH, respectively, while LLMOpt (Jiang et al., 2025) uses Kahneman-Tversky Optimization (KTO) on synthetic data. SIRL (Chen et al., 2025) combines SFT with RL to further boost results. However,

reproducibility of these works remains a challenge: some models are highly prompt-sensitive and complete training data and latest checkpoints are often unavailable.

# 4 METHOD

We now detail the complete pipeline of our framework; see Fig. 3 for an overview.

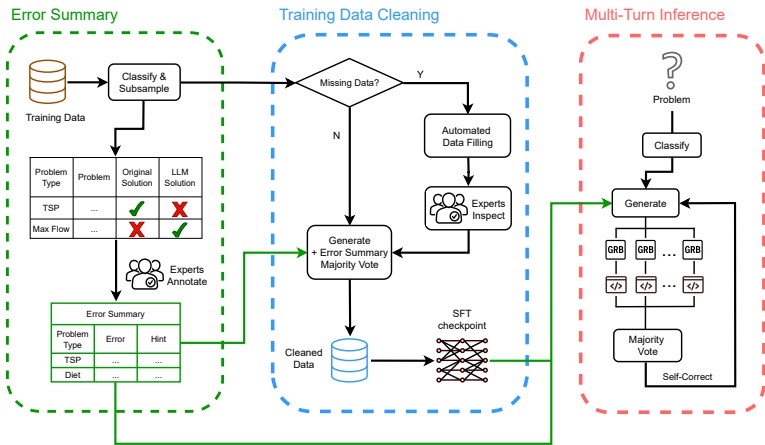

Figure 3: An overview of our training data cleaning and multi-turn inference pipeline.

## 4.1 PROBLEM-CLASS SPECIFIC ERROR ANALYSIS

While highly capable LLMs already show strong potential for optimization formulation, in this work we take on the ambitious goal of further improving their performance. Upon examining their outputs on a small training subset, we find that formulation mistakes still occur, and often similar types of mistakes recur within the same optimization category: for example, in the Traveling Salesman Problem (TSP), the LLMs often mess up the subtour elimination constraint by incorrectly applying it to the fixed starting node (Fig. 4, middle box). Similarly, we see frequent mistakes in the signs of flow-conservation constraints in network-flow and inventory management problems; for example, if $B[u]$ denotes the net supply of a node $u$ (i.e., positive if there is extra supply), the flow conservation equation should read [outflow from $u$] - [inflow into $u$] = $B[u]$, but we often found the outflow and inflow terms swapped. This suggests a simple yet intuitive intervention: *summarize short, targeted hints that capture the most common failure modes in each optimization category and attach the appropriate hint when solving problems from that category.*

To this end, we develop a class-based error analysis. We classify all problems in the open-source optimization training sets (OptMATH (Lu et al., 2025) and OR-Instruct (Tang et al., 2024), which is the training data corresponding to the IndustryOR benchmark) into the 53 seed classes defined in the OptMATH problem generators; these classes provide a wide coverage of typical MILP-type problems. For each class, we run GPT-OSS-20B to produce answers and select 10–20 instances where its answer disagrees with the original ground-truth label. This label mismatch can be attributed either to model errors in the solution generation or quality issues with the questions themselves (details in §4.2). Our optimization experts manually review each item to identify the source of the disagreement; when the model is at fault, we write a short error summary and a concrete hint that would have prevented the mistake (see Fig. 4 (left) for an example). We then aggregate these annotations into a dictionary that maps each problem type into a list of (error summary, preventive hint) pairs.

This outcome of the error analysis becomes a core component of both our training-data cleaning and our inference pipeline. Fig. 4 (right) demonstrates the power of a single hint in preventing a frequent subtour-related error. Fig. 9 (left) demonstrates the generalization capability of our hints: even without closely matched training items, the same hint family reduces mistakes on the Mamo-Complex benchmark on the biggest problem classes, yielding a 16.6% gain in overall accuracy.

| Hint Example for TSP | Response without Hint | Response with Hint |
|---|---|---|
| **Error description:** "When solving TSP, it is easy to make mistakes by not enforcing the MTZ subtour elimination constraints correctly."

**Hint:** "When solving TSP using MTZ, fix one city's position (e.g., u[0]=0) and apply u[i] - u[j] + n*x[i,j] <= n - 1 for the *remaining* cities i, j != 0 with i != j to ensure no subtours are created; do *not* apply this constraint u[i] - u[j] + n*x[i,j] <= n - 1 for the city whose position you fixed (e.g., city 0)" | **Thought:** "We need to solve TSP with 6 nodes, directed distances given (not symmetric). We need to formulate TSP as integer program: binary variables x[i,j] = 1 if travel from i to j. Constraints: each city has exactly one outgoing edge and one incoming edge. Subtour elimination: use Miller-Tucker-Zemlin (MTZ) constraints: u[i] variables continuous between 1 and n. For i != 0, u[i] >= 2, <= n. For all i!=j, u[i] - u[j] + n * x[i,j] <= n-1. Or standard: u[i] - u[j] + n*x[i,j] <= n-1. Also need to ensure start city 0? Usually we set u[0] = 0. Or we can set u[0] = 0 and enforce others between 1 and n-1. But with 6 nodes, n=6. We'll implement accordingly." | **Thought:** "We need to solve TSP with 6 nodes. Provide model: binary variables x[i,j] for i != j, objective minimize sum cost[i][j]*x[i,j]. Constraints: each node outdegree 1, indegree 1. Use MTZ subtour elimination: choose node 0 as fixed with u[0]=0, u[i] integer 1..n-1 for i>0. For all i != j, i,j > 0? Actually apply for all i != j, i>0 and j>0? The hint says not to apply constraint for city whose position fixed. So for i != j, i>0, j>0, i != j: u[i] - u[j] + n*x[i,j] <= n-1." |

Figure 4: Sample hint for a problem type (Traveling Salesman Problem, TSP) and comparison of the model's response before and after applying hints.

## 4.2 TRAINING DATA CLEANING

Building on the error analysis above, we now examine how to leverage training data to systematically improve the base model. A common approach to improve LLMs for optimization formulation is by supervised fine-tuning on the training sets attached to each benchmark. However, we find that the training data from the datasets we consider (OR-Instruct and OptMATH) exhibits quality issues that mirror, and sometimes exceed, those in the test sets. First, many training *solutions* (reasoning, code, and final answers) were synthesized by older LLMs (e.g., OR-Instruct from Tang et al. (2024) uses gpt-4 and OptMATH from Lu et al. (2025) uses DeepSeek-V3), and we often observe low-quality or internally inconsistent outputs that would propagate errors into SFT. Second, many training *questions* contain missing parameters or ambiguous phrasing, akin to the issues we documented in the benchmarks. We exemplify this issue in Appendix A.10.

Unlike test benchmarks, these training corpora are large, so exhaustive manual relabeling is impractical. We therefore design a semi-automated cleaning pipeline that combines targeted expert review with scalable LLM-assisted checks. We pursue two complementary directions in order to obtain a higher-quality training dataset suitable for learning: (i) improve *solution* quality and labels, and (ii) improve *question* quality and clarity.

**Improve solution quality.** To improve "ground-truth" quality and balance across the datasets, we:

- *Balance classes.* From the bigger dataset, OptMATH, we sample 100 training instances per problem class when available; for classes with fewer than 100 instances, we take them all.
- *Solution regeneration using error analysis and majority voting.* Here we employ a simplified version of our inference process described in more detail in §4.4. That is, we use gpt-oss-120b augmented with the class-specific error summaries and hints described in §4.1 to reduce recurrent modeling mistakes. We use majority vote with $K{=}64$ samples, yielding higher-quality solutions.
- *Filter unresolvable items.* Finally, we drop problems where neither the original code nor the regeneration process produce a valid result.

This process yields 2700 cleaned items for OR-Instruct and 2600 for OptMATH. Of these, $602/2700$ in OR-Instruct and $577/2600$ in OptMATH have answers that differ from the original labels.

**Improve question quality.** To address the issue of missing data and ambiguous description:

- *Detect and fill missing parameters.* We automate detection and completion of missing fields using the OpenAI o4-mini model, flagging $180/2700$ items in IndustryOR and $500/2600$ in OptMATH as incomplete and filling them with validated values. We then manually checked a few samples by our optimization experts to verify correctness.
- *Regenerate and clarify OptMATH instances.* Following the data generation process of OptMATH (Lu et al., 2025), we re-generate 53 instances for each seed class and back-translate the problem descriptions using the GPT-OSS-120B model. Our optimization experts manually inspect 159 re-

generated instances (three per class) to confirm that key parameters are present and consistent. Training on this regenerated data improved downstream performance compared to using the original OptMATH problems; see our ablation results in §5.3.

## 4.3 SUPERVISED FINE-TUNING

We use supervised fine-tuning (SFT) to strengthen the model's formulation and coding ability. Concretely, we fully fine-tune gpt-oss-20b on our cleaned training dataset. Let the SFT dataset be $\mathcal{D}_{\text{SFT}} = (x_i, y_i)_{i=1}^{N}$, where $x_i$ is the problem description and $y_i$ is the completion sequence formed by concatenating the model's thinking tokens, the mathematical formulation, and the solver code. We train with standard sequence-to-sequence loss: $\mathcal{L}_{\text{SFT}}(\theta) = -\mathbb{E}_{(x,y)\sim\mathcal{D}_{\text{SFT}}} \sum_{t=1}^{|y|} \log p_\theta(y_t \mid y_{<t}, x)$, where $\theta$ denotes all trainable parameters of the model; $p_\theta(\cdot)$ is the model's output distribution; $y_t$ is the target token at position $t$; and $y_{<t} := (y_1, \ldots, y_{t-1})$ are the prefix of previously generated target tokens. Full training details are provided in the Appendix A.8.

## 4.4 INFERENCE

We adopt an inference pipeline with error-aware prompting as the default, and, when additional computation budget is available, we optionally apply two test-time scaling techniques: (1) self-consistency via majority voting and (2) multi-turn correction with tool feedback. As shown in §5, these components reinforce each other to form a robust pipeline for solving optimization problems.

**Improved prompts with error-analysis-based hints.** We incorporate the class-based error analysis from §4.1 into the inference process. The model first classifies each test instance into one of the 53 classes that were defined in the training data error analysis, then augments its prompt with all error–hint pairs from that class to guide solution generation. Since the training instances from the same class can differ in their characteristics (e.g., some multi-period inventory problems allow backorders while others do not), we clarify in the prompt that hints should be applied only when relevant, allowing the model to ignore those hints that do not fit the problem description. This category-based targeted application helps the model avoid recurrent pitfalls and reduces common mistakes. On top of the hints we obtained from our analysis, we add general guidelines for correct formulation, which are derived from our optimization experts' experience.

While the hinting mechanism is our main contribution on the inference side, we also study two standard test-time scaling techniques that can further improve performance when additional computation is available.

**Self-consistency with majority voting.** Self-consistency with majority voting samples multiple reasoning traces for the same instance and returns the answer that appears most frequently, reducing sampling variance and providing modest accuracy gains at the cost of extra samples.

**Multi-turn correction.** Multi-turn correction runs a short feedback loop in which we execute the generated Python code, collect solver logs or execution errors, feed this feedback back to the model, and let it revise its formulation; with a few turns, most coding errors and some modeling issues are corrected, but each additional turn increases inference cost. Appendix A.7 provides an example of nontrivial self-corrections.

Putting it together, the full inference pipeline proceeds as follows: the initial turn classifies the problem; the next turn solves it using the returned hints for the predicted class(es); we may then generate $K$ solutions and select the majority-vote answer, and, if further compute budget is available, repeat this correction loop for $M$ rounds using tool feedback, as summarized in Alg. 1 in Appendix A.6.

## 4.5 ENSURING RELIABLE EVALUATION VIA CAREFUL TEST DATA CLEANING

We consider three of the most challenging optimization formulation benchmarks – IndustryOR, Mamo Complex, and OptMATH – where even the strongest models at the time only report accuracies up to 20%–50% (Jiang et al., 2025; Liu et al., 2025; Lu et al., 2025). By contrast, earlier LP-centric benchmarks (e.g., NL4LP (Ramamonjison et al., 2022), NLP4LP (AhmadiTeshnizi et al., 2024))

already see 90–95%+ accuracy (see e.g. (Liu et al., 2025; Lu et al., 2025)), highlighting substantial headroom on these harder tasks.

On closer inspection, however, we find a surprising fact: many errors stem not from model capability but from *issues in the benchmarks themselves*: missing or ambiguous problem data, incorrect reference answers, and rounding inconsistencies in the evaluation pipeline, resulting in 30% - 60% test instances being incorrect. Despite recent efforts to clean these benchmarks (Tang et al., 2024; Chen et al., 2025), such issues remain widespread.

To ensure the reliability of evaluation, we carefully re-cleaned the test data of these three benchmarks. This was a non-trivial process, requiring over a month of manual effort from a team of optimization experts (with experience level of Professors and PhD students in Operations Research). The errors include missing data, ambiguous problem descriptions, integral vs. fractional variables and more. See Appendix A.11 for details, including how we corrected the issues. After cleaning, we observe a remarkable gain in the accuracy across all three benchmarks. With the same gpt-oss-20b model and inference strategy, the average accuracy increases from 40%–60% on the original releases to 70%–90% on our corrected sets. We will release our fixes and annotations for each problem to make results comparable across papers and better reflect true model ability.

## 5 EXPERIMENTS

### 5.1 EXPERIMENTAL SETUP

**Datasets.** Our evaluation is performed on our rigorously cleaned versions of three challenging and widely-used benchmarks: **IndustryOR** (Tang et al., 2024), **Mamo-Complex** (Huang et al., 2024), and **OptMATH** (Lu et al., 2025), each with around 100-160 problems (see Appendix A.11 for details). We selected these datasets as they are commonly used in previous works and represent some of the most complex formulation tasks in the literature, providing a strong signal for model capabilities. While we also considered other benchmarks such as OptiBench (Yang et al., 2025b) (605 problems) and ComplexOR (Xiao et al., 2024) (18 problems), our initial experiments revealed that performance on these datasets has either saturated or does not effectively differentiate between models of varying scales; moreover, their sizes are either too small (yielding a noisy evaluation signal) or too large to feasibly clean with the same rigor. A more detailed discussion of our dataset selection and the comprehensive cleaning process is provided in Appendix A.10. For completeness, we also report results on cleaned versions of IndustryOR released by SIRL (Chen et al., 2025), LogiOR (Yang et al., 2025a), and Xiao et al. (2025a), as well as on cleaned versions of Mamo-Complex from SIRL and Xiao et al. (2025a).

**Metrics.** Our primary metric is solution accuracy, which we report at the first generation turn (Turn 1) and after five turns of iterative self-correction (Turn 5). Our prompts require an executable Python script using GurobiPy to formulate and solve the problem. Following OptMATH (Lu et al., 2025), we mark a solution as correct if the normalized error $|\hat{z} - z^\star|/(|z^\star| + 1)$ is below $10^{-6}$, where $\hat{z}$ is the model's objective value and $z^\star$ is the ground-truth objective. To extract this value, we insert a print statement emitting the optimal objective with a unique tag and parse it from the execution log, as in SIRL. We also assess the effect of self-consistency by comparing results without majority voting ($K = 1$) against results with majority voting over $K = 8$ samples, grouping answers within a relative and absolute tolerance of $10^{-6}$ to account floating-point variations. To ensure statistical robustness, all reported results are averaged across 10 independent experiments using different random seeds.

**Baselines.** To comprehensively assess our contributions, we compare OptiMind against several baselines: (1) we compare with the GPT-OSS-20B base model without any fine-tuning to quantify the gains from our training methodology; (2) at inference, we test a variant using only basic prompts without our class-specific error hints to verify the effectiveness of our domain-informed guidance; (3) we also benchmark our model against other frontier models, including GPT-O4-MINI and GPT-5 and (4) an open-source models of similar size, QWEN3-32B; (5) furthermore, we re-evaluated SIRL-GUROBI-32B (Chen et al., 2025) for direct comparison with the current state-of-the-art models in the field of optimization. Note that we were unable to replicate the reported performance of other open-source models like OptMATH and LLMOpt.

## 5.2 MAIN RESULTS

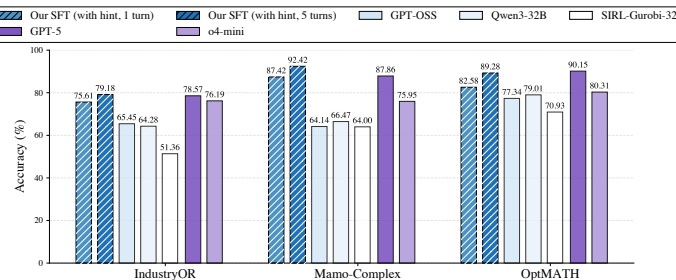

Figure 5: Average accuracy for different models without majority voting. Models colored in blue are open-sourced models under 32B parameters, and models in purple are closed-sourced frontier models. OPTIMIND outperforms open-source baselines and performs comparably to O4-MINI in the single-turn setting and to GPT-5 with multi-turn self-corrections.

We now assess the impact of OptiMind's pipeline on solution accuracy.

**Overall comparison.** Our results are visualized in Figure 5, and the full numerical values are reported in Table 1. As can be seen, OPTIMIND achieves comparable performance to the frontier close-sourced models with much larger size while clearly outperforming other open-source models of similar size. In the single-turn, no-hint regime, our SFT model consistently improves 5%-21% over the GPT-OSS-20B base model, 6%-19% over QWEN3-32B, and 12%-21% over the domain-specific state-of-the-art model SIRL-GUROBI32B across all three benchmarks. Combining SFT with error-analysis hints, our OPTIMIND framework performs comparably to, and often improves upon O4-MINI under the single-turn setting: it's within 0.6% on IndustryOR, and improves by +11.5% on Mamo-Complex, and +2.6% on OptMATH. With five turn, OPTIMIND further substantially improve upon o4-mini and perform comparably with GPT-5: it is within about 0.8% on OptMATH, and exceeds it by +0.9% and +4.6% on IndustryOR and Mamo-Complex respectively. We also observe similar trends on the cleaned versions of IndustryOR and Mamo-Complex released by prior work (Chen et al. (2025), Yang et al. (2025a), and Xiao et al. (2025a)); detailed numbers for these additional test sets are reported in Table 2 in Appendix A.1.

**Results with OptiMind's SFT training.** Our SFT training fed with the cleaned training data processed through OptiMind has demonstrable gains. Under the single-turn without hints, our SFT model outperforms the base model by +5.3% on IndustryOR, +21.4% on Mamo-Complex, and +5.3% on OptMATH. These gains remain when we apply error-analysis hints and 5-turn self-correction: with +4.62%, +5.55%, and +6.58% on IndustryOR, Mamo-Complex, and OptMATH, respectively, as shown in Figure 6. Additional ablation studies can be found in §5.3.

**Results with OptiMind's error analysis.** Additionally, using class-specific error analysis and the associated hints at inference consistently lift single-turn accuracy across models and datasets. Typical gains from "no hint" to "with hints" across the baselines and our SFT model (single-turn, no majority voting) range from +3% to +6% on IndustryOR, +2% to +10% on Mamo-Complex, and +1% to +4% on OptMATH, with rare small dips around −0.5% on OptMATH for a few models. Interestingly, even GPT-5 sees a significant improvement in accuracy when using the hints (up to +4.93% on IndustryOR), suggesting that hints can enhance the performance of even very strong models, encoding domain-specific information that seems complementary to general model's capabilities. See Fig. 7 for the detailed results.

As a deeper dive, we examine Mamo-Complex, which is dominated by five problem classes. Fig. 8 shows a per-class breakdown of GPT-OSS-20B model without and with hints, and and our SFT model with hints at inference: accuracy increases are broad and most pronounced on types prone to error related to sign conventions (e.g., within flow-conservation-type constraints) and in complex structural constraints (e.g., subtour elimination constraints in TSP), and the SFT+hint setting further improves over the base+hint model across these classes. While IndustryOR and OptMATH contain more heterogeneous types, we observe the same consistent pattern of improvement; see Appendix A.9 for details.

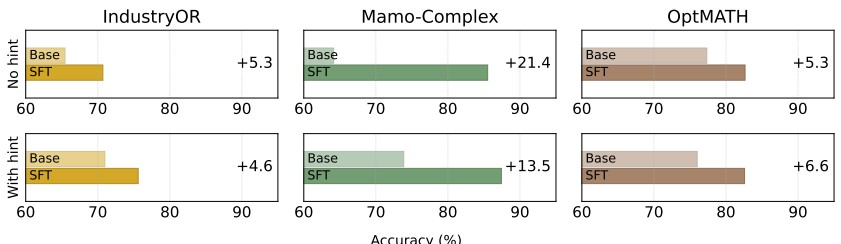

Figure 6: GPT-OSS-20B vs. our SFT model under no-hint prompting, comparing the prompts without hint and with hint under single-turn, no majority-voting setting. We report SFT improvements over the base in percentage points.

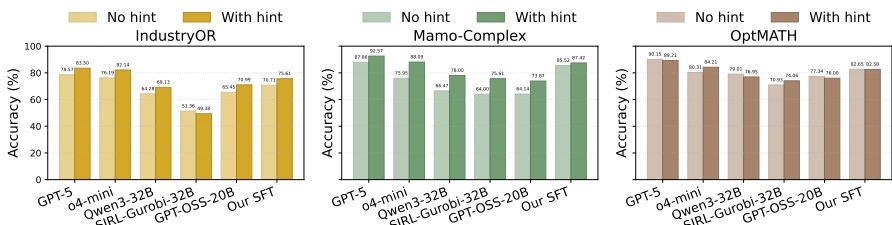

Figure 7: Single-turn accuracy of various models on three cleaned benchmarks w/ and w/o hints. We show consistent improvement in accuracy from adding error-analysis hints.

**Effect of test-time scaling**   We evaluate the effect of two test-time scaling techniques—multi-turn self-correction and majority voting (MV)—when additional computation is available. Figure 9 show that both techniques provide consistent accuracy improvements over single-turn inference for the GPT-OSS-20B base model and our SFT model, with and without hints. MV and multi-turn are complementary, but the marginal benefit of MV diminishes once multiple turns are used, and gains from additional turns also exhibit diminishing returns as compute increases. Overall, the combination of SFT and hints with a small number of self-correction turns delivers the best accuracy-compute tradeoff across benchmarks.

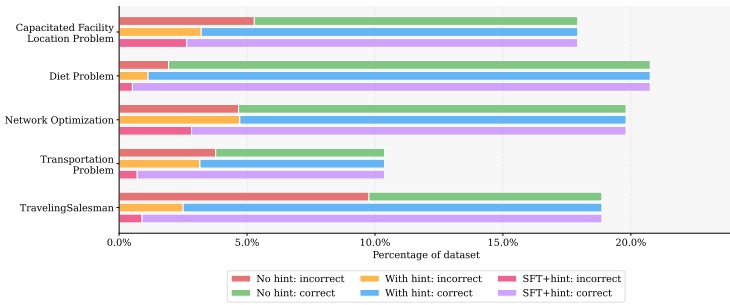

Figure 8: Accuracy by problem type on Mamo-Complex for GPT-OSS-20B (single-turn, no majority voting) and our SFT model. Infrequent problem types are omitted.

## 5.3   ADDITIONAL ABLATION STUDIES

To understand better where the gains come we perform ablation studies on the two main components of our framework – data cleaning and inference strategies.

**Ablation on training data.** To illustrate the effectiveness of our cleaned training data, we evaluate seven alternative data configurations: (1) *Original data:* the original OptMATH set plus COPTPY-to-Python/GurobiPy translations from O4-MINI, keeping only instances whose final solutions match the provided answers; (2) *No OR-Instruct fixing:* training on the original OR-INSTRUCT questions without fixing missing-data issues; (3) *No OptMATH back-translation:* training on the original

| Model | K | Hints | IndustryOR | | Mamo-Complex | | OptMATH | |
|---|---|---|---|---|---|---|---|---|
| | | | 1 turn | 5 turns | 1 turn | 5 turns | 1 turn | 5 turns |
| GPT-OSS-20B | 1 | no | $65.45 \pm 4.79$ | $75.80 \pm 3.26$ | $64.14 \pm 4.24$ | $88.29 \pm 1.26$ | $77.34 \pm 3.49$ | $88.05 \pm 1.33$ |
| | | yes | $70.99 \pm 1.84$ | $78.71 \pm 2.07$ | $73.87 \pm 2.43$ | $90.61 \pm 1.36$ | $76.00 \pm 2.24$ | $87.72 \pm 1.95$ |
| OUR SFT | 1 | no | $70.71 \pm 3.15$ | $77.55 \pm 1.18$ | $85.52 \pm 2.56$ | $91.09 \pm 1.34$ | $\mathbf{82.65 \pm 1.83}$ | $88.43 \pm 1.46$ |
| | | yes | $\mathbf{75.61 \pm 2.65}$ | $\mathbf{79.48 \pm 2.28}$ | $\mathbf{87.42 \pm 1.05}$ | $\mathbf{92.42 \pm 0.96}$ | $82.58 \pm 1.60$ | $\mathbf{89.28 \pm 0.86}$ |
| GPT-OSS-20B | 8 | no | $79.79 \pm 2.08$ | $80.40 \pm 1.34$ | $85.90 \pm 2.30$ | $88.57 \pm 1.50$ | $90.93 \pm 2.45$ | $90.23 \pm 1.99$ |
| | | yes | $82.65 \pm 2.63$ | $81.53 \pm 1.47$ | $90.84 \pm 1.44$ | $91.37 \pm 0.73$ | $91.25 \pm 0.96$ | $90.62 \pm 1.22$ |
| OUR SFT | 8 | no | $79.59 \pm 1.66$ | $81.42 \pm 1.97$ | $92.47 \pm 1.24$ | $92.95 \pm 0.62$ | $\mathbf{91.56 \pm 1.41}$ | $\mathbf{91.87 \pm 1.33}$ |
| | | yes | $\mathbf{82.86 \pm 1.91}$ | $\mathbf{83.26 \pm 1.69}$ | $\mathbf{93.23 \pm 0.58}$ | $\mathbf{93.27 \pm 0.58}$ | $89.60 \pm 1.16$ | $90.54 \pm 0.77$ |

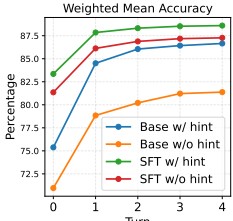

Figure 9: **Left**: Accuracies of the GPT-OSS-20B base model and our SFT model across majority voting $K$ and turn counts on the three benchmarks. **Right**: Weighted mean accuracy over five self-correcting turns on the three benchmarks (each weighted by its sample count). Results for each benchmark can be found in Appendix A.1.

OPTMATH questions without back-translation; (4) *Single-shot solution:* generating solutions without majority voting; (5) *No-hint:* removing error-analysis hints when re-generating solutions; (6) *Answer-aligned data:* keeping the original problem and final answer, generating 64 solutions with GPT-OSS-120B, and selecting the sample that matches the original answer even if that answer is imperfect; and (7) *20b data:* replacing GPT-OSS-120B with GPT-OSS-20B as the response generator for training trajectories.

Figure 11 compares these configurations under the single-turn, no-hint setting and reports the weighted average accuracy across the three benchmarks (weighted by sample count). All ablations underperform our full data recipe, which combines missing-data fixes, manual resolution of key ambiguities, OptMATH re-generation with back-translation, and GPT-OSS-120B-generated trajectories. In particular, using GPT-OSS-20B to generate data)is competitive but consistently weaker than the GPT-OSS-120B-based data, showing that our full data recipe is effective.

**Ablation on classification.** We also test the classification component on the Mamo-Complex benchmark, comparing four configurations: (1) classification performed by GPT-OSS (our default); (2) (approximate) oracle classification by human expert manually labeled classes; (3) random assignment of classes; and (4) applying all class-specific hints together in the prompt without classification. Table 3 shows that configuration (1) achieves the best performance and is even slightly better than using human-expert manually assigned classes, suggesting that the GPT-OSS-based classifier is well aligned with the error patterns from our analysis. Configurations (2) and (4) perform similarly and both substantially outperform random classification in (3), indicating that our class structure and hints capture meaningful error modes, and that targeted hint selection remains helpful and robust even when classification is imperfect.

## 6 CONCLUSION

We present *OptiMind* – a framework for formulating mixed-integer linear optimization problems with LLMs, combining a strong fine-tuned model with error-aware prompting and optional test-time scaling. Both training and inference benefit from domain-specific targeted hints that capture common error patterns within each optimization class and provide concise guidance to prevent those errors. The resulting pipeline attains strong performance across multiple benchmarks. While future LLMs may embed more expert knowledge, we believe the principles and techniques of our framework will remain essential, and we are keen to apply them to domains such as supply chain management or adapt them to enterprise-specific scenarios to drive real-world impact.

ETHICS STATEMENT

This paper does not involve human subjects, personally identifiable data, or sensitive applications. We do not foresee direct ethical risks. We follow the ICLR Code of Ethics and affirm that all aspects of this research comply with the principles of fairness, transparency, and integrity.

REPRODUCIBILITY STATEMENT

We ensure reproducibility on both theoretical and empirical fronts. For theory, we include all formal assumptions, definitions, and complete proofs in the appendix. For experiments, we describe model architectures, datasets, preprocessing steps, hyperparameters, and training details in the main text and appendix.

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

# A APPENDIX

## CONTENTS

## A.1 ADDITIONAL EVALUATION RESULTS

Table 1: Accuracies of all models on expert-cleaned benchmarks in the single-sample setting (majority $K = 1$) under different hint configurations.

| Model | Hints | IndustryOR | | Mamo-Complex | | OptMATH | |
|---|---|---|---|---|---|---|---|
| | | 1 turn | 5 turns | 1 turn | 5 turns | 1 turn | 5 turns |
| GPT-5 | no | $78.57 \pm 3.09$ | $81.12 \pm 1.79$ | $87.86 \pm 1.55$ | $89.36 \pm 1.92$ | $90.15 \pm 0.69$ | $92.18 \pm 0.78$ |
| | yes | $83.50 \pm 2.84$ | $85.20 \pm 1.25$ | $92.57 \pm 1.19$ | $93.42 \pm 0.62$ | $89.21 \pm 1.78$ | $90.31 \pm 1.04$ |
| O4-MINI | no | $76.19 \pm 1.05$ | $82.14 \pm 2.11$ | $75.95 \pm 2.38$ | $87.78 \pm 1.15$ | $80.31 \pm 2.02$ | $87.18 \pm 0.89$ |
| | yes | $82.14 \pm 2.47$ | $85.88 \pm 1.19$ | $88.09 \pm 2.02$ | $91.67 \pm 1.52$ | $84.21 \pm 2.17$ | $90.46 \pm 1.16$ |
| QWEN3-32B | no | $64.28 \pm 3.00$ | $69.39 \pm 3.00$ | $66.47 \pm 1.64$ | $77.14 \pm 3.18$ | $79.01 \pm 2.41$ | $79.49 \pm 1.84$ |
| | yes | $69.13 \pm 2.00$ | $74.23 \pm 3.85$ | $78.00 \pm 1.73$ | $84.76 \pm 2.12$ | $76.95 \pm 1.35$ | $80.85 \pm 1.35$ |
| SIRL-GUROBI32B | no | $51.36 \pm 1.39$ | $55.27 \pm 1.63$ | $64.00 \pm 1.41$ | $72.67 \pm 1.37$ | $70.93 \pm 1.94$ | $75.46 \pm 1.62$ |
| | yes | $49.38 \pm 3.02$ | $53.06 \pm 3.61$ | $75.61 \pm 2.05$ | $78.57 \pm 0.95$ | $74.06 \pm 1.39$ | $76.71 \pm 2.30$ |
| GPT-OSS-20B | no | $65.45 \pm 4.79$ | $75.80 \pm 3.26$ | $64.14 \pm 4.24$ | $88.29 \pm 1.26$ | $77.34 \pm 3.49$ | $88.05 \pm 1.33$ |
| | yes | $70.99 \pm 1.84$ | $78.71 \pm 2.07$ | $73.87 \pm 2.43$ | $90.61 \pm 1.36$ | $76.00 \pm 2.24$ | $87.72 \pm 1.95$ |
| OUR SFT | no | $70.71 \pm 3.15$ | $77.55 \pm 1.18$ | $85.52 \pm 2.56$ | $91.09 \pm 1.34$ | $\mathbf{82.65 \pm 1.83}$ | $88.43 \pm 1.46$ |
| | yes | $\mathbf{75.61 \pm 2.65}$ | $\mathbf{79.48 \pm 2.28}$ | $\mathbf{87.42 \pm 1.05}$ | $\mathbf{92.42 \pm 0.96}$ | $82.58 \pm 1.60$ | $\mathbf{89.28 \pm 0.86}$ |

Table 2: Accuracies under different base models and hint settings on benchmarks cleaned by parallel works. SIRL stands for Chen et al. (2025), LogiOR stands for Yang et al. (2025a), and Survey stands for Xiao et al. (2025a).

| Model | K | Hints | IndustryOR-SIRL | | IndustryOR-LogiOR | | IndustryOR-Survey | | MAMO-Complex-SIRL | | MAMO-Complex-Survey | |
|---|---|---|---|---|---|---|---|---|---|---|---|---|
| | | | 1 turn | 5 turns | 1 turn | 5 turns | 1 turn | 5 turns | 1 turn | 5 turns | 1 turn | 5 turns |
| GPT-5 | 1 | no | 69.67 | 71 | 68.29 | 70.12 | 66.67 | 68.25 | 72.57 | 74.05 | 65.76 | 66.21 |
| | | yes | 72 | 73 | 76.82 | 78.04 | 70.23 | 69.04 | 88.17 | 89.65 | 85.58 | 86.48 |
| O4-MINI | 1 | no | 61.33 | 64.67 | 66.26 | 74.79 | 68.25 | 69.84 | 57.96 | 69.95 | 58.55 | 66.67 |
| | | yes | 68.67 | 70 | 71.54 | 74.39 | 66.67 | 63.49 | 81.77 | 86.53 | 78.82 | 86.48 |
| SIRL-GUROBI32B | 1 | no | 29.25 | 29.87 | 38.9 | 39 | 47.85 | 47.85 | 53.05 | 53.1 | 83.06 | 82.97 |
| | | yes | 30.8 | 31.2 | 35.1 | 35.6 | 46.67 | 46.67 | 65.56 | 65.61 | 77.02 | 77.2 |
| GPT-OSS-20B | 1 | no | 54.3 | 62.7 | 59.87 | 65.97 | 62.85 | 67.38 | 60.39 | 77.83 | 68.1 | 78.82 |
| | | yes | 57.3 | 63.3 | 66.09 | 70 | 62.38 | 67.38 | 73.59 | 85.27 | 77.47 | 85.85 |
| OUR SFT | 1 | no | 57.3 | 63.7 | 63.71 | 67.68 | 67.61 | 68.57 | 79.8 | 85.32 | 85.22 | 85.94 |
| | | yes | 59.2 | 62.6 | 67.68 | 71.64 | 69.04 | 71.42 | 84.43 | 87.68 | 87.38 | 90.09 |

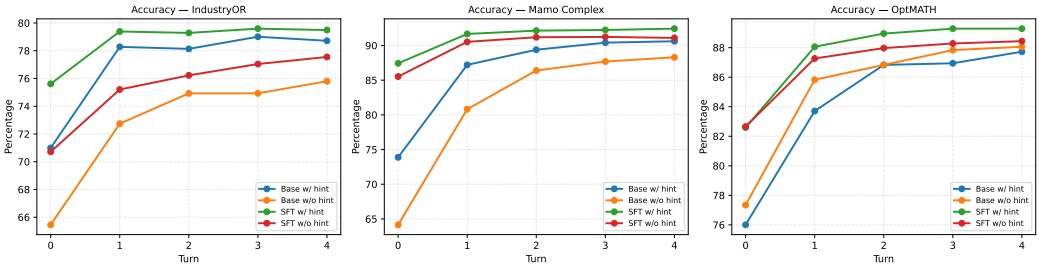

Figure 10: Accuracy over five self-correcting turns on the three benchmarks.

## A.2 DETAILS OF ABLATION STUDIES

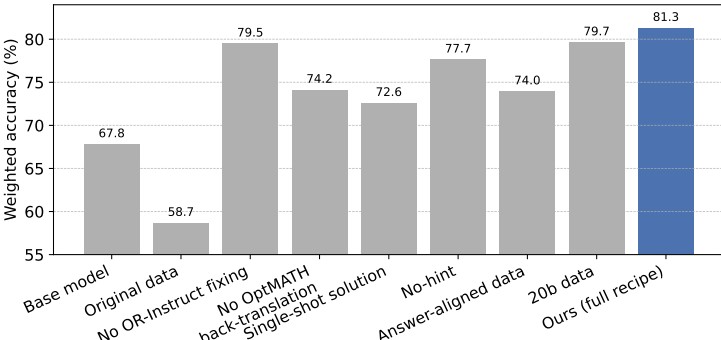

Figure 11: Ablation on training-data configurations. We compare seven variants of our training data recipe under the single-turn, no-hint setting and report weighted average accuracy across IndustryOR, Mamo-Complex, and OptMATH.

Table 3: Ablation on classification and hinting strategies on Mamo-Complex. We report single-turn and 5-turn accuracies (in percentage points).

| Classification / Hint Strategy | 1 turn | 5 turn |
|---|---|---|
| GPT-OSS classifier (ours) | 83.81 | 89.90 |
| Human expert manual classification | 81.04 | 89.61 |
| All hints, no classification | 78.57 | 89.23 |
| Random classes | 72.57 | 87.23 |
| No hints | 70.00 | 86.57 |

## A.3 Sample Problem Description and Output Optimization Model

Below we have a sample problem description from the dataset OptMATH:

---

**Question:** Maximize the total profit by determining the optimal production, inventory, and sales plan for three products (Product 0, Product 1, and Product 2) over a five-period planning horizon. The profit per unit sold is $269 for Product 0, $282 for Product 1, and $241 for Product 2. The holding cost for each unit stored is $15 per period. At the start of the planning horizon (Period 0), the production of each product must equal the sum of its sales and inventory for that period. For each subsequent period, the inventory carried over from the previous period plus the production in the current period must equal the sum of sales and inventory for the current period. The total production time required for all products on each machine type must not exceed the available capacity. The capacities are 480 hours per period for grinders, 320 hours per period for drills, and 160 hours per period for borers. Each unit of Product 0 requires 1 hour on grinders, 1 hour on drills, and 2 hours on borers. Each unit of Product 1 requires 1 hour on grinders, 1 hour on drills, and 2 hours on borers. Each unit of Product 2 requires 1 hour on grinders, 2 hours on drills, and 1 hour on borers. The inventory of each product at the end of any period cannot exceed 80 units. The maximum number of units that can be sold for each product in each period is as follows: Product 0: 48 in Period 0, 43 in Period 1, 58 in Period 2, 58 in Period 3, and 61 in Period 4; Product 1: 54 in Period 0, 56 in Period 1, 45 in Period 2, 46 in Period 3, and 40 in Period 4; Product 2: 57 in Period 0, 52 in Period 1, 68 in Period 2, 40 in Period 3, and 60 in Period 4. At the end of the planning horizon (Period 4), the inventory of each product must be exactly 20 units. Ensure that the final inventory levels meet the target requirements and that production does not exceed machine capacities.

\# Note:
- The Code must include:```python

import gurobipy as gp
from gurobipy import GRB
```
- Make sure the model variable is named 'model'.
- Avoid using "<" >" in Gurobi constraints; instead, use "<=" or ">=" as appropriate.
- Carefully determine whether the variable is an integer or a continuous variable.

---

Below we have the ground-truth output code formulates this problem as an MILP; the output also includes an initial description of the mathematical formulation in markdown format, which is omited for brevity:

```python
import gurobipy as gp
from gurobipy import GRB

# Data
products = [0, 1, 2]
periods = [0, 1, 2, 3, 4]
machines = ['grinders', 'drills', 'borers']

profit = {0: 269, 1: 282, 2: 241}
holding_cost = 15

machine_capacity = {
    'grinders': 480,
    'drills': 320,
    'borers': 160
}

machine_time = {
    'grinders': {0: 1, 1: 1, 2: 1},
    'drills': {0: 1, 1: 1, 2: 2},
    'borers': {0: 2, 1: 2, 2: 1}
}

max_sales = {
    0: {0: 48, 1: 43, 2: 58, 3: 58, 4: 61},
    1: {0: 54, 1: 56, 2: 45, 3: 46, 4: 40},
    2: {0: 57, 1: 52, 2: 68, 3: 40, 4: 60}
}
```

```python
# Model
model = gp.Model("Production_Planning")

# Decision Variables
x = model.addVars(products, periods, name="Production")
s = model.addVars(products, periods, name="Sales")
I = model.addVars(products, periods, name="Inventory")

# Objective Function
model.setObjective(
    gp.quicksum(profit[i] * s[i, t] for i in products for t in periods) -
    gp.quicksum(holding_cost * I[i, t] for i in products for t in periods),
    GRB.MAXIMIZE
)

# Constraints
# Initial Inventory Balance (Period 0)
model.addConstrs((x[i, 0] == s[i, 0] + I[i, 0] for i in products), name="Initial_Balance")

# Inventory Balance (Periods 1-4)
model.addConstrs((I[i, t-1] + x[i, t] == s[i, t] + I[i, t] for i in products for t in periods
↪   if t >= 1), name="Inventory_Balance")

# Machine Capacity Constraints
model.addConstrs((gp.quicksum(machine_time[m][i] * x[i, t] for i in products) <=
↪   machine_capacity[m] for m in machines for t in periods), name="Machine_Capacity")

# Sales Constraints
model.addConstrs((s[i, t] <= max_sales[i][t] for i in products for t in periods),
↪   name="Sales_Constraint")

# Inventory Constraints
model.addConstrs((I[i, t] <= 80 for i in products for t in periods),
↪   name="Inventory_Constraint")

# Final Inventory Target
model.addConstrs((I[i, 4] == 20 for i in products), name="Final_Inventory_Target")

# Optimize
model.optimize()

# Results Interpretation
if model.status == GRB.OPTIMAL:
    print("Optimal Solution Found!")
    print(f"Total Profit: ${model.ObjVal:.2f}")

    for i in products:
        print(f"\nProduct {i}:")
        for t in periods:
            print(f"Period {t}: Production={x[i,t].X:.2f}, Sales={s[i,t].X:.2f},
            ↪   Inventory={I[i,t].X:.2f}")
else:
    print("No optimal solution found.")
```

## A.4 PROBLEM CLASSIFICATION

For problem classification, we use all 49 seed classes from the OptMATH dataset, finding that these categories are diverse and cover a comprehensive proportion of classes in our training set. Furthermore, we obtain corresponding natural language examples for each problem category from the OptMATH repository. Lu et al. (2025); Table 4 contains the 49 seed class names, and Table 5 shows examples of corresponding examples. When prompting the LLM to classify the problem, we provide all problem categories and natural language examples to assist the language model.

---

**Problem Classification Prompt**

```
You are a helpful Assistant with expertise in mathematical modeling
and the Gurobi solver.  You will be given a natural language question.
Please classify the question into one or more of the following
categories:
{list of categories}

If the question does not fit any of the above categories, please
provide a label for the Category that best describes the problem type.

Category examples:
{natual language example of each category}

Please classify the following question into one or more categories:
{question}
Return only the categories in a python list, without any additional
text.
```

---

| | | |
|---|---|---|
| Aircraft Assignment | Aircraft Landing | Bin Packing |
| Blending Problem | Capacitated Facility Location Problem | Capacitated Lot-sizing Problem (CLSP) |
| Car Selection Assignment | Contract Allocation | Diet Problem |
| Factory Planning Problem | Flow Shop Scheduling | Job Shop |
| Knapsack | Multicommodity Capacitated Network Design | MarketShare |
| Set Multi-Cover | PortfolioOptimization | Revenue Management Problem |
| Assignment Problem | Set Cover | Discrete Lot-sizing and Scheduling Problem |
| Static Line Planning | Structure Based Assignment Problem | SupplyChain |
| TravelingSalesman | Facility Dispersion Problem | Military Personnel Deployment Problem |
| Production Planning Problem | Facility Dispersion Problem | Network Optimization |
| Lot-Sizing Problem | Operations Optimization | Capacitated Vehicle Routing Problem with Time Windows (CVRPTW) |
| Facility Location Problem | Cutting Stock Problem | Unit Commitment Problem |
| Farm Planning | Transportation, Airline Industry, Resource Allocation | Multi-Commodity Transportation Problem |
| Minimum Cost Flow Problem | Assignment Problem | Multi-Commodity Network Flow Problem |
| Transportation Problem | Profit Maximization Problem | Revenue Maximization Problem |
| Facility Location Problem Transportation Problem | Production Planning Problem | Team Formulation Problem |

Table 4: Problem categories obtained from OptMATH

| Subclass | Natural Language Example |
|---|---|
| Knapsack | A hiker is preparing for a 3-day outdoor hiking trip. They need to select the most valuable combination of equipment and supplies from many available options within the limited backpack capacity. The items include:
* Tent: weight 5kg, value 90 points
* Sleeping bag: weight 3kg, value 80 points* Portable stove: weight 1kg, value 60 points
* Drinking water: weight 2kg, value 70 points
* Dry food: weight 1.5kg, value 65 points
* First aid kit: weight 0.5kg, value 50 points
Assuming the backpack has a maximum weight capacity of 10kg, the hiker's goal is to select the combination of items with the highest total value while not exceeding the weight limit. Each item must be either taken in its entirety or left behind; partial items cannot be taken. |
| Team Formulation Problem | As a team leader, you have several projects requiring different expertise levels in different areas. You want to assign a set of people to these projects. Each person has a level of expertise in each of the areas. How to assign people to projects to minimize the maximum skill shortage from the project expertise requirement? |
| Transportation Problem | A company has three warehouses (origins) with supplies of 100, 150, and 200 units, respectively. There are four retail stores (destinations) with demands of 80, 120, 90, and 160 units, respectively. The cost of transporting one unit from each warehouse to each retail store is given. The goal is to determine the optimal transportation plan that minimizes the total transportation cost while meeting all supply and demand constraints. |

Table 5: Examples of OptMATH classes and their corresponding natural language examples. Full data can be found in `https://github.com/optsuite/OptMATH/tree/main/data/generators`
.

## A.5    PROMPTS FOR AUTOMATICALLY REPAIRING MISSING DATA

We design an automatic procedure to fill in missing data in the public available training datasets ORLM Tang et al. (2024) and OptMATH Lu et al. (2025). First, we generate a gurobipy code to each incomplete question by prompting OpenAI's reasoning models (o1, o3, o4-mini), which often attempts to fill in synthetic data to complete the question. We then extract the synthesized data from the code which corresponds to the missing values. We then use the following prompt to ask o4-mini to produce a modified question with the missing data filled in, where we provide both the original question and the gurobipy code as inputs. Lastly, we manually inspect a subset of 100 repaired questions from OptMATH to make sure that the missing data are successfully filled in.

---

**Prompt for Missing Data Infilling**

```
You are given a optimization question and its mathematical solution as
well as python code.  The question may have missing data or parameters
that are filled in by the solution.  Your task is to identify if the
solution introduces new numbers/values that should be added into the
question.
Question:  {question}
Solution:  {completion}

If the question has missing data and the solution provides specific
numbers that fills these missing parameters, return the modified
question with the missing parameters imputed.  Do not modify anything
else except for adding in missing data.  Otherwise, return NO MISSING
DATA. Only return the modified question or NO MISSING DATA, nothing
else.
```

---

Below is an example question from OptMATH that has missing data infilled by our prompting. The red texts are from the original question with missing data. The green texts are the infilled texts.

---

**Example of Automatic Missing Data Infilling**

```
# Question: Aircraft Assignment for Route Coverage

You are responsible for managing the assignment of aircraft to various
routes for an airline company. The goal is to minimize the total
operational costs while ensuring that all route demands are met
and that the availability of each aircraft type is not exceeded.

#### Aircraft and Routes:
- There are 6 types of aircraft (aircraft_0 to aircraft_5) available
for assignment.
- There are 7 routes (route_0 to route_6) that need to be serviced.

#### Aircraft Availability:
Each aircraft type has a limited number of units available for
assignment:
- Aircraft_0: 4 units available
- Aircraft_1: 5 units available
- Aircraft_2: 4 units available
- Aircraft_3: 4 units available
- Aircraft_4: 5 units available
- Aircraft_5: 5 units available

#### Route Demands:
Each route has a specific demand that must be satisfied by the combined
capabilities of the assigned aircraft. The demands are as follows:
- Route_0: Requires at least 275 units of capacity
- Route_1: Requires at least 213 units of capacity
- Route_2: Requires at least 228 units of capacity
- Route_3: Requires at least 265 units of capacity
```

```
  - Route_4: Requires at least 226 units of capacity
  - Route_5: Requires at least 277 units of capacity
  - Route_6: Requires at least 246 units of capacity

  #### Aircraft Capabilities:
  Each aircraft type contributes differently to the capacity of each
  route. For example:
  - **Aircraft_0** contributes 100 units to Route_0, 101 units to Route_1,
  88 units to Route_2, and so on.
  - **Aircraft_1** contributes 80 units to Route_0, 88 units to Route_1,
  111 units to Route_2, and so on.
  - Similar contributions are defined for all other aircraft types across
  all routes.

  #### Aircraft Capabilities:
  Each aircraft type contributes differently to the capacity of each
  route. The capacity contributions p_ij (aircraft i
  to route j) are:

  - Aircraft_0: [100, 101, 88, 95, 110, 120, 105]
  - Aircraft_1: [80, 88, 111, 90, 85, 95, 100]
  - Aircraft_2: [120, 110, 105, 115, 125, 130, 110]
  - Aircraft_3: [90, 95, 100, 105, 110, 115, 120]
  - Aircraft_4: [110, 120, 115, 125, 130, 135, 140]
  - Aircraft_5: [95, 100, 105, 110, 115, 120, 125]

  #### Operational Costs:
  Assigning an aircraft to a route incurs a specific cost. For example:
  - Assigning **Aircraft_0** to **Route_0** costs 3778 units.
  - Assigning **Aircraft_0** to **Route_1** costs 3344 units.
  - Assigning **Aircraft_1** to **Route_0** costs 3660 units.
  - Similar costs are defined for all other aircraft-route combinations.

  #### Operational Costs:
  Assigning an aircraft to a route incurs a specific cost c_ij
  (aircraft i to route j):

  - Aircraft_0: [3778, 3344, 3555, 3666, 3777, 3888, 3999]
  - Aircraft_1: [3660, 3444, 3555, 3666, 3777, 3888, 3999]
  - Aircraft_2: [4000, 3888, 3777, 3666, 3555, 3444, 3333]
  - Aircraft_3: [3555, 3666, 3777, 3888, 3999, 4000, 4111]
  - Aircraft_4: [3888, 3777, 3666, 3555, 3444, 3333, 3222]
  - Aircraft_5: [3666, 3555, 3444, 3333, 3222, 3111, 3000]

  #### Objective:
  Determine the number of each aircraft type to assign to each route
  such that:
  1. The total operational cost is minimized.
  2. The total capacity provided by the assigned aircraft meets
  or exceeds the demand for each route.
  3. The number of aircraft assigned does not exceed the availability of
  any aircraft type.

  #### Constraints:
  - The total number of aircraft of each type assigned across all routes
  must not exceed its availability.
  - The combined capacity of all aircraft assigned to a route must meet
  or exceed the route's demand.
  - The number of aircraft assigned to any route must be a non-negative
  integer.
```

---

**Algorithm 1** MULTITURNINFERENCEWITHMAJORITYVOTING

---

**Require:** problem instance $Q$, error-analysis library $H$ (maps problem type $\rightarrow$ list of (error, hint) pairs), LLM Generator $G$, majority voting number $K$, number of correction rounds $M$.

    $t \leftarrow \text{CLASSIFYTYPE}_G(Q)$                                      ▷ predict problem type

    $\text{hints} \leftarrow H[t]$                                ▷ retrieve hints of corresponding type

    $\left(s_1^{(0)}, \cdots s_K^{(0)}\right) \leftarrow \text{FIRSTTURNGENERATE}_G(Q, \text{hints}, K)$            ▷ generate $K$ solutions

    $a^{(0)}, \text{stdout}^{(0)}, \text{stderr}^{(0)} \leftarrow \text{GETMAJORITYRESULTS}\left(s_1^{(0)}, \cdots s_K^{(0)}\right)$

    **for** $m = 1$ to $M - 1$ **do**

        $\left(s_1^{(m)}, \cdots s_K^{(m)}\right) \leftarrow \text{SELFCORRECTIONGENERATE}_G(Q, \text{stdout}^{(m-1)}, \text{stderr}^{(m-1)}, K)$    ▷ get $K$ self-correction responses

        $a^{(m)}, \text{stdout}^{(m)}, \text{stderr}^{(m)} \leftarrow \text{GETMAJORITYRESULTS}\left(s_1^{(m)}, \cdots s_K^{(m)}\right)$

    **end for**

    **return** $\hat{a}^{(M-1)}$                                      ▷ Return the final result

---

**Algorithm 2** GETMAJORITYRESULTS$(s_1, \cdots s_K)$

---

**Require:** Solution trajectories $s_1, \cdots s_K$

    **for** $k = 1$ to $K$ **do**

        $\left(a_k, \text{stdout}_k, \text{stderr}_k\right) \leftarrow \text{EXTRACTANSWER}(s_k)$    ▷ Extract code and get system output for each solution

    **end for**

    $\hat{k} \leftarrow \text{GETMAJORITYVOTEINDEX}\left(a_1, \cdots, a_K\right)$

    **return** $\left(a_{\hat{k}}, \text{stdout}_{\hat{k}}, \text{stderr}_{\hat{k}}\right)$

---

## A.6 MULTI-TURN INFERENCE PROMPTS

We use the following prompts during our multi-turn inference. At the first turn, we prompt the LLM with the question and asks for the corresponding gurobipy code. Optionally, we may include an error-analysis hint for the problem's category, which we find generally improve the performance. Algorithm 1 shows a pseudocode of our inference strategy.

---

**First Turn User Prompt (Not Using Error Analysis Hint)**

```
You are an expert in optimization and mixed integer programming.  You
are given an optimization problem and you need to solve it using
gurobipy.

{question}

Reason step by step before generating the gurobipy code.
When you respond, first think carefully.
After thinking, output the math modeling of the problem.
Finally output a ```python ...``` code block that solves the problem.

The code must include:
import gurobipy as gp
from gurobipy import GRB
```

---

**First Turn User Prompt (Using Error Analysis Hint)**

```
You are an expert in optimization and mixed integer programming.  You
are given an optimization problem and you need to solve it using
gurobipy.

{question}
```

```
Below are hints for avoiding common mistakes often seen for this
problem type.  Avoid them if applicable.

Error analysis #1:  {Error Summary 1}, {Hint 1}
...
Error analysis #k:  {Error Summary k}, {Hint k}

Instructions for applying error-analysis hints:
- Review the provided hints and identify which ones are applicable to
this problem.
- Please apply ALL applicable hints.
- Before applying any hint or writing constraints, check the sign and
direction of every variable and coefficient for consistency (e.g.,
profit = revenue - cost; capacities as <= flow conservation as =).

General modeling checklist (follow rigorously):
- Units:  Use correct units everywhere and ensure the objective's
units match the goal (e.g., dollars for cost/profit, distance for TSP,
time for scheduling).  Do not mix units (e.g., minutes with hours,
dollars with 1000 dollars) without converting.

Reason step by step before generating the gurobipy code.
When you respond, first think carefully.
After thinking, output the math modeling of the problem.
Finally output a '''python ...''' code block that solves the problem.

The code must include:
import gurobipy as gp
from gurobipy import GRB
```

From the LLM's response, we extract and execute the gurobipy code to obtain standard output
(STDOUT) and standard error (STDERR). Our multi-turn pipeline then feeds this execution feed-
back back to the LLM, prompting it to self-correct its previous solution, as shown below.

**Self-Correction Feedback Prompt**

```
Running the given gurobipy code block result in the following standard
output and error.
Please analyze the standard output and error and provide your thoughts
on the correctness of the code and the output, and if it is not
correct, provide a corrected version of the code.

The standard output and error message (if any) for your generated code
are as follows:

[STDOUT] {Standard Output of the gurobipy program (gurobipy log)}

[STDERR] {Standard Error of the gurobipy program (if exists)}

Based on the output and error messa ge (if any), please think step by
step in the <think> ...  </think> block to determine if the code is
correct.

Based on your thoughts, please provide an (improved) gurobipy code in
the '''python\n''' code block if you have identified mistakes in the
previous code.

If the previous code is correct, you do not need to provide a new
gurobi code block again.  Your performance will be evaluated based on
the correctness of the final '''python\n''' code block.
```

```
# Note:
- The Code must include:```python import gurobipy as gp from gurobipy
import GRB ```
- Make sure the model variable is named `model`.
- Avoid using "<" and ">" in Gurobi constraints; instead, use "<=" or
">=".
- Carefully determine whether the variable is an integer or
continuous.
```

## A.7 EXAMPLE OF THE EFFECT OF MULTI-TURN INFERENCE

We describe an example where multi-turn inference corrected the formulation.

This instance is production optimization problem from the **IndustryOR** dataset. In this instance, a factory has a warehouse for holding inventory with capacity of 15,000 cubic meters. In addition, it can use an external warehouse (at an additional cost).

The first turn of the inference produced a model with the **incorrect** constraint "e[t] <= vol - wh_vol_cap", where e[t] is a decision variable choosing how many cubic meters of external warehouse to use in period $t$, vol denotes the total volume of (internal + external) warehouse used, and wh_vol_cap $= 15,000$ is the internal warehouse capacity. Notice that this constraint is incorrect because it allows the volume used vol to be bigger than the capacity wh_vol_cap while having external warehouse usage e[t] = 0. The correct inequality is in the opposite direction, namely "e[t] >= vol - wh_vol_cap", which forces that the external warehouse usage to cover for any excess over the internal warehouse capacity.

The second turn of the inference used the output from the first turn (in particular, the first-turn formulation and the optimal decisions of this first model) to detect this mistake and come up with the correct constraint. Here is the relevant snipped from the second-turn reasoning:

---

**Thought of the second turn of inference (snippet)**

Volume per month: compute volume each month. July: 0.2*45k
+ 0.4*15k = 9000 + 6000 = 15000 cubic meters. That equals
capacity. So no external. August: 0.2*75k + 0.4*0 = 15000.
OK. September: 0.2*127.5k + 0.4*0 = 25500. **Exceeds 15000 by
10500**. But model forced e_t <= vol - wh_vol_cap. So e_t would
be 10500. But output shows e_t 0. So maybe e_t not used? Let's
check: The constraint e[t] <= vol - wh_vol_cap. For September
vol=25500, wh_vol_cap=15000 => RHS=10500. So e_t <= 10500. But
objective includes 0.5*e_t. To minimize cost, e_t should be
0. But we didn't enforce e_t >= vol - wh_vol_cap? No, we only
enforced e_t <= vol - wh_vol_cap. That allows e_t to be 0 even if
vol > wh_vol_cap. But we need to enforce that external volume equals
excess volume. **We used inequality incorrectly**. We should enforce
e_t >= vol - wh_vol_cap, not <=.

---

Table 6: Generation / evaluation sampling parameters.

| Parameter | Value |
| --- | --- |
| Batch size | 16 |
| Learning rate | 2e-5 |
| Max prompt length | 16384 |
| Max response length | 16384 |
| Prompt length truncation | left |
| Max epochs | 3 |

Table 7: Generation / evaluation sampling parameters.

| Parameter | Value |
| --- | --- |
| Temperature | 0.9 |
| Top p | 0.95 |
| Max tokens | 60000 |

## A.8 EXPERIMENT DETAILS

We performed Supervised fine-tuning (SFT) on a single node with eight NVIDIA HGX B200 GPUs. All solution generation in the data-cleaning pipeline and evaluation tasks were run on 4 compute nodes, each with eight 80GB NVIDIA H100 GPUs. We use GPT-OSS-20B as the base model and adapt the VERL framework for SFT. Concretely, we train from the UNSLOTH/OPENAI-GPT-OSS-20B-BFLOAT16 variant to avoid MXFP4 precision pitfalls and to leverage Unsloth's compatibility fixes for gpt-oss, and uses FSDP2 strategy to distribute training across the eight GPUs. We tune the learning rate over $[10^{-6}, 4 \times 10^{-5}]$ via a grid search, and select the best configuration on the validation split under the single-turn, no-hint setting. Key SFT hyperparameters and optimizer settings are listed in Table 6.

For solution generation in data cleaning, we host OPENAI/GPT-OSS-20B with VLLM. For evaluation, we serve all SFT checkpoints and the BF16 base (UNSLOTH/OPENAI-GPT-OSS-20B-BFLOAT16) via SGLANG, since VLLM did not support our Unsloth-adapted variant when we conducted the evaluation. We use top-p decoding for all generation and inference. We tune the sampling temperature over $[0.5, 1.0]$ via a grid search, and select the best configuration on the validation split under the single-turn, no-hint setting. The common sampling settings for our SFT model are summarized in Table 7.

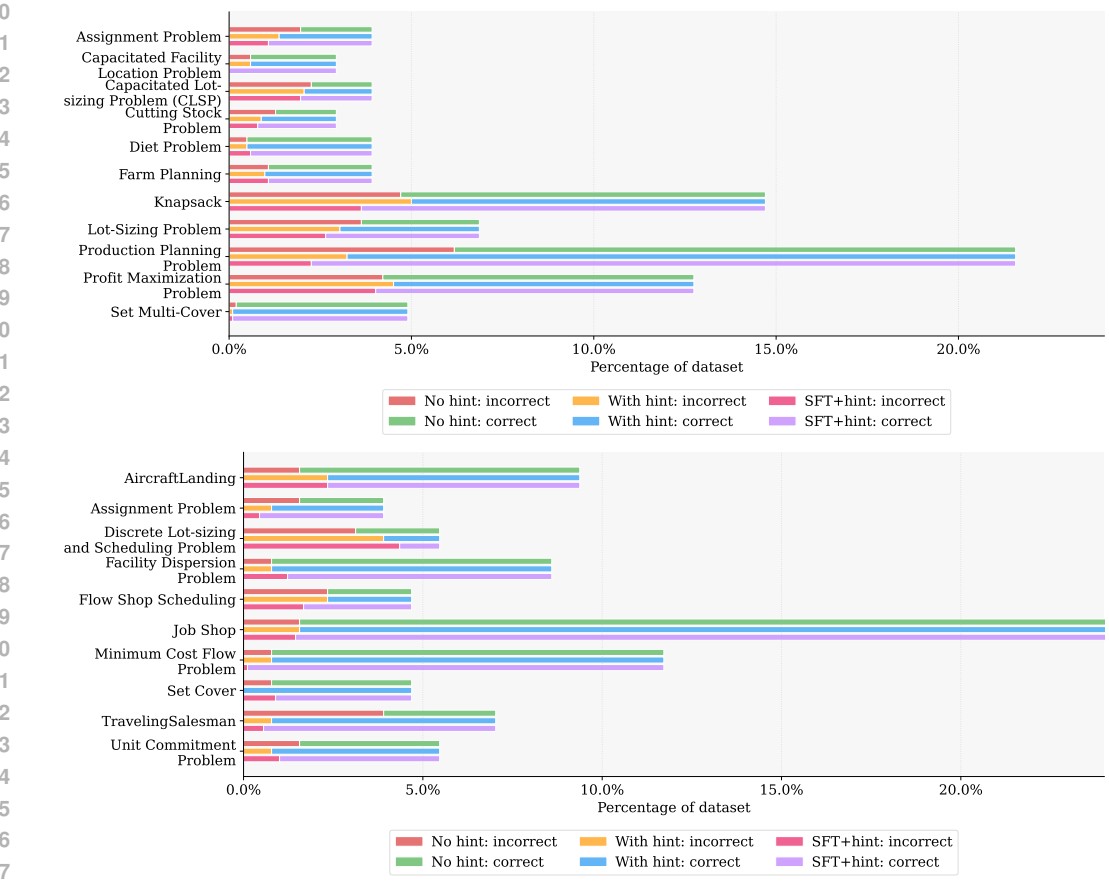

Figure 12: Accuracy by problem type on IndustryOR (top) and OPTMath (bottom), measured via single-turn evaluation with GPT-OSS-20B (no majority voting). Problem types occurring in less than 2.5% of instances are omitted in the figure.

## A.9 DETAILS OF ERROR-ANALYSIS HINTS AT INFERENCE

Figure 13 presents a per-problem-class breakdown of the impact of using error-analysis and associated hints at inference on the GPT-OSS-20B model.

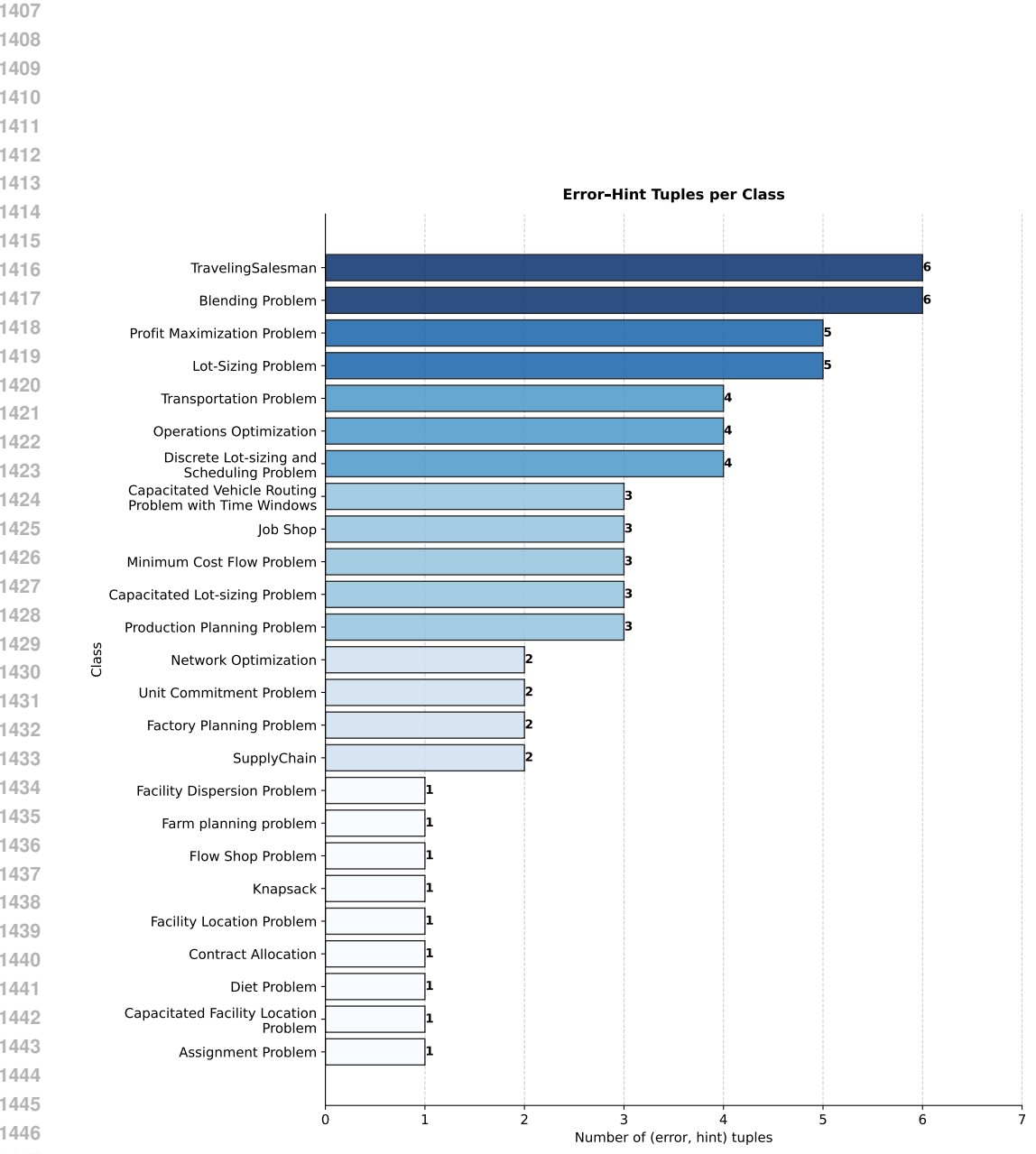

Figure 13: Number of error-hint pairs that experts collected per class.

## A.10   DETAILS OF TRAINING DATA CLEANING

Below we provide an example of cleaning training instance of the Job Shop problem with missing data. We describe the semi-automated procedure for infilling missing data in Appendix A.5.

---

**Job Shop Problem (Before)**

```
# Question:In a manufacturing facility, six distinct jobs must be processed on a set
of machines.  Each job consists of two operations, and each operation must be performed
on a specific machine.  The goal is to schedule these operations to minimize the total
completion time, known as the makespan.  Each operation has a processing time of 1 unit,
and operations within the same job must follow a specific sequence, with at least 1 unit
of time between consecutive operations.  Each machine can process only one operation
at a time, and the order of operations on shared machines is determined by binary
decision variables.  These binary variables ensure that no two operations overlap on
the same machine.  The makespan must be at least as large as the completion time of every
operation, and all start times must be non-negative.  The binary variables take values
of 0 or 1, and a large constant (100,000) is used in the machine capacity constraints to
enforce the order of operations.  The objective is to determine the start times of all
operations and the order in which they are processed on each machine, ensuring that all
constraints are satisfied and the makespan is minimized.

# Note:
- The Code must include:```python
import gurobipy as gp
from gurobipy import GRB
```
- Make sure the model variable is named 'model'.
- Avoid using "<" and ">" in Gurobi constraints; instead, use "<=" or ">=" as
appropriate.
- Carefully determine whether the variable is an integer or a continuous variable.
```

---

**Job Shop Problem (After)**

```
# Question:In a manufacturing facility, six distinct jobs must be processed on a set of
machines.  Each job consists of two operations, and each operation must be performed on a
specific machine.  In particular, for Job 0:  Operation 0 is assigned to Machine 2, and
Operation 1 to Machine 1, for Job 1:  Operation 0 is assigned to Machine 0, and Operation
1 to Machine 2, for Job 2:  Operation 0 is assigned to Machine 1, and Operation 1 to
Machine 0, for Job 3:  Operation 0 is assigned to Machine 2, and Operation 1 to Machine 2,
for Job 4:  Operation 0 is assigned to Machine 1, and Operation 1 to Machine 0, and for
Job 5:  Operation 0 is assigned to Machine 0, and Operation 1 to Machine 1.  The goal is
to schedule these operations to minimize the total completion time, known as the makespan.
Each operation has a processing time of 1 unit, and operations within the same job must
follow a specific sequence, with at least 1 unit of time between consecutive operations.
Each machine can process only one operation at a time, and the order of operations on
shared machines is determined by binary decision variables.  These binary variables ensure
that no two operations overlap on the same machine.  The makespan must be at least as
large as the completion time of every operation, and all start times must be non-negative.
The binary variables take values of 0 or 1, and a large constant (1000) is used in the
machine capacity constraints to enforce the order of operations.  The objective is to
determine the start times of all operations and the order in which they are processed on
each machine, ensuring that all constraints are satisfied and the makespan is minimized.

# Note:
- The Code must include:```python
import gurobipy as gp
from gurobipy import GRB
```
- Make sure the model variable is named 'model'.
- Avoid using "<" and ">" in Gurobi constraints; instead, use "<=" or ">=" as
appropriate.
- Carefully determine whether the variable is an integer or a continuous variable.
```

Table 8: Representative incorrect problems and fixes in IndustryOR and OptMATH.

| Benchmark | Problem Index | Optimization Class | Why the original problem is incorrect | How we fixed it |
|---|---|---|---|---|
| IndustryOR | 24 | – | The problem did not state an objective function. | We added a well-specified objective (minimize pollution). |
| IndustryOR | 46 | – | The original instance was infeasible but had a ground-truth objective value of 472.3. | We updated the data to make the instance feasible and adjusted the ground-truth objective accordingly. |
| OptMATH | 35 | Facility Location Problem (FLP) | The problem describes selecting towers to cover regions, but the description is missing data about which towers cover which regions. | We generated the missing coverage data semi-automatically and then verified it manually. |
| OptMATH | 36 | Job Shop | The instance omits the number of machines and specifies only a partial assignment of jobs to machines. | We generated the missing machine and assignment data semi-automatically and then verified it manually. |

Table 9: High-level breakdown of benchmark errors. Some instances belong to multiple error categories. Percentages are relative to the total number of problems in each benchmark.

| Benchmark | # (%) Incorrect | Missing Data | Ambiguous | Infeasible | Wrong GT | Underspecified | Specific Solver | Non-linear | Inconsistent | Integral / Fractional |
|---|---|---|---|---|---|---|---|---|---|---|
| IndustryOR | 50 (50%) | 0 (0%) | 11 (11%) | 2 (2%) | 25 (25%) | 5 (5%) | 5 (5%) | 1 (1%) | 2 (2%) | 4 (4%) |
| OptMATH | 68 (41%) | 48 (29%) | 11 (7%) | 1 (0.6%) | 10 (6%) | 1 (0.6%) | 0 (0%) | 0 (0%) | 0 (0%) | 0 (0%) |

## A.11 DETAILS OF BENCHMARKS CLEANING

This section describes representative issues we observed in the benchmark data. These fall into six main categories: missing data, ambiguous problem descriptions, integral vs. fractional variables, wrong solutions, infeasible problems, and out-of-scope problems. We next provide additional details and examples that highlight these challenges.

All cleaned benchmarks have been manually verified by domain experts. Despite earlier claims of correctness in prior work, we found substantial residual error rates (approximately 30–50%) in the original IndustryOR and OptMATH test sets. We apply a structured and repeatable cleaning process with multiple expert passes to address these issues and document every modification in the released correction logs.

Table 8 shows representative examples of incorrect problems and their fixes in IndustryOR and OptMATH, while Table 9 summarizes the frequency and distribution of major error types across both benchmarks. A complete, per-instance correction record is available in the HTML summary tables described in our supplementary material.

Cleaning these benchmarks is nontrivial given the high initial error rate in the original data sources. To ensure rigor, we conduct a total of four passes of expert review. All corrections are documented in the supplementary material, and we plan to release the full cleaning logs on GitHub for broader visibility.

### A.11.1 MISSING DATA

We observe that many problems contain missing values for key parameters. For example, an assignment problem in OptMATH reads *"For example, assigning aircraft_0 to route_0 costs 2552 units, to route_1 costs 4340 units, and so on"* without providing the additional costs. Interestingly, when data is missing, the OpenAI o-series models synthesize reasonable values for the missing parameters during the formulation generation process. We therefore manually identify problems with missing data, and leverage the fabricated values in their solutions to fill them back into the question description, followed by manual inspection to validate correctness.

### A.11.2 AMBIGUOUS PROBLEM DESCRIPTIONS

We also observe that many problems contain ambiguities or inconsistencies. For example, a facility location problem in IndustryOR had the following objective function *"to achieve the goal of min-*

*imizing costs and maximizing coverage area"*, thus containing two conflicting objectives. These issues are corrected via manual inspection.

### A.11.3 INTEGRAL VS FRACTIONAL VARIABLES

Many problems exhibit ambiguity regarding whether decision variables are integer or continuous. For example, a production optimization problem in IndustryOR refers to "units of products", while the reported ground truth corresponds to the production of fractional products. We resolve this by updating the ground truth value, adding explicit clarification sentences that enforce integrality or continuity.

---

**Example of Integer vs. Fractional**

```
A company produces product A and product B. Each unit of product A sold generates a profit
of £30, while each unit of product B sold generates a profit of £10.  The company can
allocate a maximum of 40 hours per week for production.  Producing one unit of product A
requires 6 hours, while producing one unit of product B requires 3 hours.  Market demand
requires that the quantity of product B produced must be at least three times the quantity
of product A. The storage space occupied by product A is four times that of product B, and
a maximum of four units of product A can be stored per week.

Formulate a model for this problem.

Reported ground truth:  146.667
Corrected ground truth:  160
```

---

### A.11.4 WRONG SOLUTIONS

The benchmarks also contained problems where the provided ground truth value was incorrect. For instance, some minimum-cost flow problems in the OptMATH dataset reported wrong optimal costs (e.g., one problem reported optimal cost of 6 as the ground truth, while the correct optimal value was 4). These errors were all corrected with manual inspection by optimization experts.

---

**Example of Wrong Solution**

```
You are responsible for managing the distribution of emergency medical supplies across
eight cities in a region.  Each city has a specific supply of medical resources and a
demand that must be met to ensure adequate healthcare coverage.  The goal is to minimize
the total transportation cost while ensuring that all cities receive the necessary
supplies and that no distribution routes exceed their capacity.

City Supply and Demand:
- **City 0** has a net demand of 1 unit of medical supplies.
- **City 1** has a balanced supply and demand (net demand of 0 units).
- **City 2** has a net supply of 1 unit of medical supplies.
- **City 3** has a net demand of 2 units of medical supplies.
- **City 4** has a balanced supply and demand (net demand of 0 units).
- **City 5** has a balanced supply and demand (net demand of 0 units).
- **City 6** has a net supply of 2 units of medical supplies.
- **City 7** has a balanced supply and demand (net demand of 0 units).

Transportation Costs:  The cost of transporting medical supplies between cities varies
depending on the route.  Below are the transportation costs per unit of supplies:

- **From City 0**:  To City 1 costs 3, to City 2 costs 2, to City 3 costs 2, to City 4
costs 2, to City 5 costs 3, to City 6 costs 3, and to City 7 costs 1.  - **From City 1**:
To City 0 costs 1, to City 2 costs 2, to City 3 costs 3, to City 4 costs 1, to City 5
costs 2, to City 6 costs 1, and to City 7 costs 2.  - **From City 2**:  To City 0 costs
2, to City 1 costs 2, to City 3 costs 3, to City 4 costs 3, to City 5 costs 2, to City 6
costs 1, and to City 7 costs 2.  - **From City 3**:  To City 0 costs 1, to City 1 costs 2,
to City 2 costs 1, to City 4 costs 3, to City 5 costs 3, to City 6 costs 2, and to City
7 costs 3.  - **From City 4**:  To City 0 costs 3, to City 1 costs 2, to City 2 costs 1,
to City 3 costs 1, to City 5 costs 3, to City 6 costs 2, and to City 7 costs 2.  - **From
City 5**:  To City 0 costs 1, to City 1 costs 2, to City 2 costs 1, to City 3 costs 2, to
City 4 costs 1, to City 6 costs 2, and to City 7 costs 1.  - **From City 6**:  To City 0
costs 2, to City 1 costs 3, to City 2 costs 1, to City 3 costs 1, to City 4 costs 1, to
City 5 costs 1, and to City 7 costs 1.  - **From City 7**:  To City 0 costs 1, to City 1
costs 1, to City 2 costs 3, to City 3 costs 1, to City 4 costs 2, to City 5 costs 3, and
to City 6 costs 2.
```

```
Route Capacity Constraints:  Each route between cities has a maximum capacity for
transporting medical supplies:

- **From City 0**:  To City 1 (7 units), to City 2 (7 units), to City 3 (7 units), to
City 4 (7 units), to City 5 (8 units), to City 6 (8 units), and to City 7 (8 units).  -
**From City 1**:  To City 0 (8 units), to City 2 (7 units), to City 3 (8 units), to City
4 (8 units), to City 5 (7 units), to City 6 (7 units), and to City 7 (9 units).  - **From
City 2**:  To City 0 (8 units), to City 1 (7 units), to City 3 (7 units), to City 4 (7
units), to City 5 (7 units), to City 6 (9 units), and to City 7 (7 units).  - **From City
3**:  To City 0 (7 units), to City 1 (7 units), to City 2 (9 units), to City 4 (8 units),
to City 5 (7 units), to City 6 (7 units), and to City 7 (9 units).  - **From City 4**:
To City 0 (9 units), to City 1 (7 units), to City 2 (8 units), to City 3 (9 units), to
City 5 (7 units), to City 6 (7 units), and to City 7 (7 units).  - **From City 5**:  To
City 0 (7 units), to City 1 (8 units), to City 2 (9 units), to City 3 (9 units), to City
4 (8 units), to City 6 (9 units), and to City 7 (8 units).  - **From City 6**:  To City
0 (9 units), to City 1 (8 units), to City 2 (7 units), to City 3 (8 units), to City 4
(8 units), to City 5 (7 units), and to City 7 (8 units).  - **From City 7**:  To City 0
(9 units), to City 1 (8 units), to City 2 (7 units), to City 3 (9 units), to City 4 (9
units), to City 5 (8 units), and to City 6 (8 units).

City Capacity Constraints:  Each city has a maximum capacity for receiving medical
supplies:

- **City 0**:  Can receive up to 19 units.
- **City 1**:  Can receive up to 15 units.
- **City 2**:  Can receive up to 15 units.
- **City 3**:  Can receive up to 14 units.
- **City 4**:  Can receive up to 15 units.
- **City 5**:  Can receive up to 15 units.
- **City 6**:  Can receive up to 14 units.
- **City 7**:  Can receive up to 16 units.

Objective:  Your task is to determine the optimal distribution of medical supplies between
the cities to minimize the total transportation cost while ensuring that all cities meet
their supply and demand requirements, no route exceeds its capacity, and no city exceeds
its receiving capacity.

Reported ground truth:  6.
Correct ground truth:  4.
```

### A.11.5   INFEASIBLE PROBLEMS

Beyond problems with wrong solutions, we also observe problems that are infeasible. An example
is provided below. We fix these problems by appropriately updating the data so that the problem
admits a feasible solution.

### Example of Infeasible Problem

```
"A university computer lab hires 4 undergraduates (designated 1, 2, 3, and 4) and 2
graduate students (designated 5 and 6) for duty answering questions.  The maximum duty
hours from Monday to Friday and the hourly wage for each person are shown in Table 5-9.

Table 5-9
Student ID,Wage (CNY/h),Monday,Tuesday,Wednesday,Thursday,Friday
1,10.0,6,0,6,0,7
2,10.0,0,6,0,6,0
3,9.9,4,8,3,0,5
4,9.8,5,5,6,0,4
5,10.8,3,0,4,8,0
6,11.3,0,6,0,6,3

The lab operates from 8:00 AM to 10:00 PM, and there must be one and only one student
on duty during open hours.  It is also required that each undergraduate must work at
least 8 hours per week, and each graduate student must work at least 7 hours per week.
Additionally, supplement the following requirements:  each student can work no more than 2
shifts per week, and no more than 3 students can be scheduled for duty each day.  Based on
these conditions, establish a new mathematical model."
```

### Example of Fixed Infeasible Problem

```
"A university computer lab hires 4 undergraduates (designated 1, 2, 3, and 4) and 2
graduate students (designated 5 and 6) for duty answering questions.  The maximum duty
```

```
hours from Monday to Friday and the hourly wage for each person are shown in Table 5-9.

Table 5-9
Student ID | Wage (CNY/h) | Monday | Tuesday | Wednesday | Thursday | Friday
1 | 10.0 | 6 | 0 | 6 | 0 | 7
2 | 10.0 | 0 | 6 | 0 | 6 | 7
3 | 9.9 | 4 | 8 | 4 | 0 | 5
4 | 9.8 | 5 | 5 | 6 | 0 | 4
5 | 10.8 | 4 | 0 | 4 | 8 | 0
6 | 11.3 | 5 | 6 | 0 | 6 | 3

The lab operates from 8:00 AM to 10:00 PM, and there must be one and only one student
on duty during open hours.  It is also required that each undergraduate must work at
least 8 hours per week, and each graduate student must work at least 7 hours per week.
Additionally, each student can work no more than 2 shifts per week, and no more than 3
students can be scheduled for duty each day.

Based on these conditions, establish a mathematical model to determine the work schedule
that satisfies all requirements."
```

### A.11.6 OUT-OF-SCOPE PROBLEMS

Finally, we also observe a small fraction of non-linear problems. For instance, OptMATH contains certain second-order cone programming and quadratically constrained programming problems. We deliberately omit these non-linear instances from the benchmarks for the following reasons:

First, all second-order test problems in the benchmark are essentially copies of the same synthetic template (see the example below). In addition, these instances suffer from severe missing-data issues: many problem statements mention, for example, the OptMATH problems with index 142, 148, 150 and 154 mentioned "12 linear constraints" but only specify parameter values for one or two constraints, so the problems are incomplete and require substantial manual reconstruction by human labelers before they can be meaningfully used.

Second, when we do run our models on these nonlinear instances, we find that the errors are very concentrated: the solver typically encodes $\|y_k\|_2 \leq t_k^2$ constraints with $t_k$ unrestricted, making the feasible region a nonconvex union of two cones. This is essentially modeling issue that we believe would be straightforward to fix if similar second-order training examples (with correctly encoded SOCs and linking equations) were available. However, **we do not find any comparable second-order problems of this type in our training data**. This means we cannot derive targeted, class-specific hints from the training corpus to address these errors, in contrast to the MILP classes where our method is designed to operate.

The closest problem family we see in the available datasets is classical mean–variance portfolio optimization with a quadratic objective and linear constraints, e.g.

$$\min_w \quad \sum_{i=1}^{n}\sum_{j=1}^{n} w_i w_j \sigma_{ij} \quad \text{s.t.} \quad \sum_{i=1}^{n} w_i r_i \geq R, \quad \sum_{i=1}^{n} w_i = 1, \qquad l_i \leq w_i \leq u_i \, \forall i.$$

whose description and structure are quite different from the synthetic second-order cone instances in the test set. In practice, hints or error patterns derived from such portfolio problems do not transfer to the benchmark's second-order templates.

Because we build upon publicly available datasets in which second-order problems are extremely scarce and not representative of the nonlinear test instances, we view a thorough treatment of these problems as out of scope for this work. That said, our framework is general: given a sufficiently rich set of similar nonlinear problems in the training data, we expect that the same error-analysis-and-hints pipeline could be applied to derive effective hints and substantially reduce these errors as well.

**Example of Non-Linear Problem**

```
Minimize the objective function with cost coefficients -0.1919592146476727 for x[0],
-1.473647303839492 for x[1], and 2.304735407761341 for x[2].  The problem is subject
to three strict equality constraints:  (1) -1.635895473616174 x[0] - 0.3973211001807447
x[1] + 0.9471007364101932 x[2] must equal -1.151224267166901, labeled as linear_eq[0];
(2) -0.5249535603075616 x[0] + 0.3668073807989349 x[1] - 0.7858336216136411 x[2] must
```

```
equal 0.2111440590441619, labeled as linear_eq[1]; and (3) -1.91474276776438 x[0] -
0.0172618223950067 x[1] - 0.4534063203075578 x[2] must equal -0.0637133383926026, labeled
as linear_eq[2].

Additionally, four auxiliary variables y_0[0], y_0[1], y_0[2], and y_0[3] are introduced,
each subject to linear constraints:  (1) -0.1884394252535835 x[0] + 0.0177104249784969
x[1] + 0.0986130842848287 x[2] + y_0[0] must equal 0.0043955531724021, labeled as R3; (2)
-0.0708818331240574 x[0] - 0.0215819660876361 x[1] - 0.0616654799104094 x[2] + y_0[1] must
equal 0.0879019749446116, labeled as R4; (3) -0.0907143884234421 x[0] + 0.0750697315409076
x[1] - 0.1169779589661307 x[2] + y_0[2] must equal 0.0357726532121181, labeled as R5; and
(4) -0.1820309484664378 x[0] + 0.0261698752429371 x[1] + 0.0667294286775777 x[2] + y_0[3]
must equal 0.0118088378651355, labeled as R6.

A decision variable t_0[0] is introduced, subject to the constraint -0.2396501315769039
x[0] + 0.3255209569765619 x[1] + 0.2624492403208943 x[2] + t_0[0] must equal
1.588464205354944, labeled as R7.  A second-order cone constraint labeled as qc0 is
imposed, ensuring that the sum of the squares of y_0[0], y_0[1], y_0[2], and y_0[3] does
not exceed the square of t_0[0], expressed as - t_0[0]^2 + y_0[0]^2 + y_0[1]^2 + y_0[2]^2
+ y_0[3]^2 <= 0.  All decision variables, including x[0], x[1], x[2], y_0[0], y_0[1],
y_0[2], y_0[3], and t_0[0], are free to take any real value.  The goal is to determine the
optimal values for these variables to minimize the objective function while satisfying all
constraints.  This is a Second-Order Cone Programming (SOCP) problem."
```

## SUPPLEMENTARY MATERIAL: RELEASED CLEANED TEST SETS

We release the expert–cleaned test sets used in our evaluation as supplementary material:

- `optimind_cleaned_industryor.csv` (99 items),

- `optimind_cleaned_mamo_complex.csv` (210 items), and

- `optimind_cleaned_optmath.csv` (130 items),

Each CSV is UTF–8 encoded and contains two columns: `question` (the complete natural–language problem statement) and `answer` (the ground–truth optimal objective value after cleaning).

The `answer` field is either (i) a single numeric value, or (ii) a JSON array of numeric values when multiple interpretations are reasonable (e.g., integer vs. fractional formulations) and thus multiple objectives are accepted. During evaluation, a prediction is considered correct if it matches *any* provided value within absolute and relative tolerances of $10^{-6}$.

These files reflect our corrections for missing parameters, ambiguity, infeasibility, wrong reference answers, and scope mismatches identified during manual review. The sets are intended strictly for *evaluation* (not training) to ensure comparability across studies. We will update the archive if community feedback identifies additional issues; any changes will be documented with version notes.

## SUPPLEMENTARY MATERIAL: DATASET CORRECTION TABLES

In addition to the cleaned CSV test sets, for the IndustryOR and OptMATH benchmarks we provide HTML summary tables that list each correction and the reason for the change. Each HTML file contains a table with columns [Problem Index, Original Problem, Original Answer, Updated Problem, Updated Answer, How did we fix it]. Figure 14 shows a screenshot of one such table, illustrating the layout and content of the HTML files. We also release comparison files `industryOR_original_vs_ours.html` and `OptMATH_original_vs_ours.html`, which align the original instances with our cleaned versions. In addition, `compare_SIRL_Ours.html` compares our cleaned IndustryOR set against the SIRL-cleaned version: although the latter was reported as corrected, we identify five instances that still exhibit issues and document them in this file. Together, these tables offer a clear, human-readable view of the structure and details of our dataset corrections.

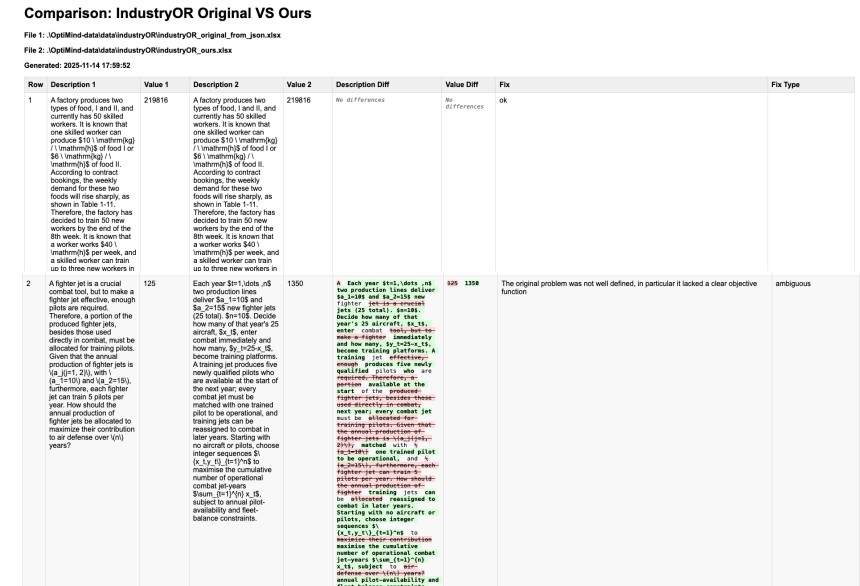

Figure 14: A screenshot of the HTML correction table in our supplementary material showing our edits for the IndustryOR benchmark.

## LLM USAGE DISCLOSURE

LLMs were used only to polish language, such as grammar and wording. These models did not contribute to idea creation or writing, and the authors take full responsibility for this paper's content.

