# OpenReview forum: "OptiMind: Teaching LLMs to Think Like Optimization Experts"
_ICLR.cc/2026/Conference — Submitted to ICLR 2026_

### Official Review · Reviewer_3yxj · 2025-10-31

**Soundness:** 2
**Presentation:** 2
**Contribution:** 2
**Rating:** 4
**Confidence:** 4

**Summary:**

This paper propose a training and inference pipline, specifically,

* **Training Data Cleaning**: A semi-automated pipeline to clean and enhance training datasets (OR-Instruct and OptMATH) by correcting errors, filling missing data, and regenerating high-quality solutions using LLMs and majority voting.

* **Multi-Turn Inference Pipeline**: An iterative inference process that incorporates:

    * Class-specific error hints.

    * Self-consistency via majority voting.

    * Solver feedback for iterative self-correction.

**Strengths:**

1. Comprehensive, End-to-End Methodology
The paper's major strength is its holistic approach, systematically improving the entire pipeline instead of just one component. It rigorously cleans both training and test data to fix widespread errors, then uses the insights from this cleaning (common error patterns) to power a sophisticated inference process. This creates a synergistic system where high-quality data and domain-informed inference work together for maximum effect.

2. Innovative and Transferable Use of Domain Knowledge
The "class-based error analysis" is a highly effective innovation. Instead of just using more data, the authors distill expert knowledge into specific "error-hint" pairs for different problem types (e.g., TSP, network flow). Injecting these targeted hints into prompts significantly boosts accuracy across various models, proving this method is a powerful and transferable blueprint for embedding expert knowledge into LLMs.

**Weaknesses:**

1.  **Limited Conceptual Novelty:** The paper's core components are incremental combinations of existing techniques rather than fundamentally new inventions. The approach of cleaning training data, using error-specific hints, and employing multi-turn inference with solver feedback is a logical and powerful integration of established ideas (like chain-of-thought, self-consistency, and tool-use) applied to a new domain. However, it does not introduce a novel algorithmic breakthrough, instead demonstrating that a meticulous, engineering-heavy application of current paradigms yields significant gains.

2.  **Significant Manual Effort and Scalability Concerns:** The framework relies heavily on manual, expert-led processes that do not scale easily. The class-based error analysis required experts to manually identify common mistakes, and the benchmark cleaning took "over a month of manual effort from a team of optimization experts." This makes the method difficult to adapt to new or broader problem domains (like nonlinear optimization) without a similar investment of expensive expert time, limiting its general applicability.

3.  **Narrow Problem Scope:** The work is exclusively focused on Mixed-Integer Linear Programs (MILPs), explicitly omitting more complex problem types like nonlinear or second-order cone programs.

**Questions:**

same as the weaknesses.

---

> ### Author Response · Authors · 2025-11-26
> **Official Comment by Authors (1/3)**
>
> **We thank the reviewer for their constructive feedback. Below, we address the specific concerns and questions raised, highlighting significant improvements in our experimental results and clarifying our methodological contributions. We further refer the reviewer to the general response where we discuss 1) the significant improvements in our experimental results, 2) justification our rigorous dataset cleaning, and 3) summary of our paper updates.**
>
> ### Response to Weakness
>
> > **Weakness 1:** Limited conceptual novelty
>
> We respectfully argue that our main contribution lies in the **systematic integration of human expert knowledge** to guide the entire lifecycle of the model, yielding gains that individual components cannot achieve in isolation:
>
> - **Data Cleaning:** We developed a structured pipeline to fix training data errors, which serves as the foundation of our performance. Many previous works have relied on synthetic data for training, yet few explicitly address the influence of synthetic data quality or propose solutions for improving large-scale synthetic datasets. Our ablation studies (Section 5.3, Figure 11) confirm that this specific combination of cleaning steps is necessary: training on our fully cleaned data ("full recipe") achieves a weighted average accuracy of 81.3%. In contrast, training on "Original data" yields only 67.8%, and even using "Answer-aligned data" only reaches 74%. This highlights that that rigorous cleaning and verification of synthetic data significantly boosts model accuracy and robustness compared to models trained on unrefined synthetic datasets.
>
> - **SFT Training:** We fine-tuned our model on this high-quality, cleaned data, resulting in substantial improvements over the underlying architecture. Our SFT model consistently improves 5%-21% over the GPT-OSS-20B base model across all three benchmarks in the single-turn setting. For instance, on the Mamo-Complex benchmark, our SFT increases accuracy from 64.14% (Base) to 85.52% (SFT), demonstrating that our data effectively unlocks domain-specific reasoning capabilities.
>
> - **Inference:** We utilize class-specific hints derived from expert analysis. Importantly, this mechanism is universally effective: adding hints consistently improves performance across all evaluated models, not just our own. This includes frontier models like GPT-5, which sees an accuracy boost of approximately +5% on IndustryOR (from 78.57% to 83.50%) and +4.7% on Mamo-Complex when using our hints. This confirms that our domain-informed hints provide complementary value even to the strongest existing models.
>
> Our ablation study (Figure 11) proves that this structured approach is crucial: the "full recipe" (81.3%) significantly outperforms training on "Original data" (67.8%) and even using "answer-aligned data" (74%), demonstrating that our expert-guided data cleaning pipeline is effective and essential for maximizing SFT performance.
>
> Furthermore, our updated experiments in the updated paper and General Response 1 demonstrate substantial improvements in model accuracy. Specifically, we achieve superior results compared to models of similar scale, narrowing the gap with large close-sourced models like GPT-5 while remaining more cost-effective. We believe that these improvements confirm the robustness and effectiveness of our approach.

---

> > ### Author Response · Authors · 2025-11-26
> > **Official Comment by Authors (2/3)**
> >
> > > **Weakness 2:** Significant manual effort and scalability concerns
> >
> > In our data cleaning pipeline, human experts were only required to manually review and correct a small subset of problems, about 200 in total. After this initial phase, our automated process scaled up by cleaning approximately 5,000 problems without further human intervention. If new data becomes available, our pipeline will be able to clean it automatically, without any further human intervention. The domain-specific hints we employ are based on 53 canonical classes; our operations research experts have confirmed that they encompass the majority of common optimization problems. While our pipeline is capable of categorizing problems into these classes autonomously, it was essential that basic per-class knowledge be provided by human experts and incorporated into our system to enable highly-accurate results.
> >
> > As our 53 predefined classes are adopted from the OptMATH seed classes, it is indeed a great question whether they cover a sufficiently wide range of problems and whether they can generalize to other benchmark datasets.
> >
> > In our implementation, we explicitly allow the GPT-OSS classifier to propose a new class when none of the existing 53 labels is appropriate. For example, for some second-order problems in the original OptMATH dataset that do not resemble any of the seed classes, the classifier labels them as “second-order optimization problem,” a new class that does not appear in the original OptMATH taxonomy. For such newly created classes, we currently do not provide any hint template during inference. Excluding the second-order optimization problems that we deliberately discard from our main experiments, fewer than 1% of benchmark instances fall outside the training classes, so this limitation has negligible impact on our present results.
> >
> > In Sec 5.3 and Table 3 in Appendix A.2 (copied below), we compare the MAMO-Complex performance with our gpt-oss based classification against several alternatives, including human expert manual classification, which we use as an approximate ground truth oracle to compare the performance.
> >
> > We see that our gpt-oss based automatic classification even outperforms human-expert manual classification in terms of achieving higher final accuracy performance, and hence we believe it serves as a good signal for validating the effectiveness of our classification method.
> >
> > Based on our discussion with OR experts, we believe the 53 classes seem to cover a wide range of optimization problems (especially for the existing benchmarks), and hence we adopt them in this current work. However, we do acknowledge that, if there are problems that fall completely out of scope from all these 53 classes (which has less than 1% in the benchmarks), our SFT model or the hint-based analysis may not be effective. While we have not detected this to be a practical issue in our current experiments, we view it as natural future work to further extend the class coverage and improve generalization.
> >
> > *Note that we do not have exact ground truth class for each benchmark problem exactly because each problem may not fall exactly into one class, and hence we need to resort to human expert to label based on their best judgement.*
> >
> > | Classification / Hint Strategy | 1 turn | 5 turn |
> > |--------------------------------------|:------:|:------:|
> > | GPT-OSS classifier (ours) | 83.81 | 89.90 |
> > | Human expert manual classification | 81.04 | 89.61 |
> > | All hints, no classification | 78.57 | 89.23 |
> > | Random classes | 72.57 | 87.23 |
> > | No hints | 70.00 | 86.57 |

---

> > > ### Author Response · Authors · 2025-11-26
> > > **Official Comment by Authors (3/3)**
> > >
> > > > **Weakness 3:** Narrow problem scope
> > >
> > > Our decision to focus on MILPs and to exclude the few nonlinear second-order instances from our main results is driven by both dataset quality and representativeness.
> > >
> > > First, all second-order test problems in the benchmark are essentially copies of the same synthetic template. For example, a typical instance asks to *“determine the optimal levels of five resources (x[0])–(x[4]) and auxiliary variables (y_i) and (t_i) to minimize a linear cost subject to 18 linear and second-order cone constraints”*, with every coefficient explicitly enumerated. In addition, these instances suffer from severe missing-data issues: many problem statements mention, for example, the OptMATH problems with index 142, 148, 150 and 154 mentioned “12 linear constraints” but only specify parameter values for one or two constraints, so the problems are incomplete and require substantial manual reconstruction by human labelers before they can be meaningfully used.
> > >
> > > Second, when we do run our models on these nonlinear instances, we find that the errors are very concentrated: the solver typically encodes $\|y_k\|_2 \leq t_k^2 $ constraints with $t_k$ unrestricted, making the feasible region a nonconvex union of two cones. This is essentially modeling issue that we believe would be straightforward to fix if similar second-order training examples (with correctly encoded SOCs and linking equations) were available. However, **we do not find any comparable second-order problems of this type in our training data**. This means we cannot derive targeted, class-specific hints from the training corpus to address these errors, in contrast to the MILP classes where our method is designed to operate.
> > >
> > > The closest problem family we see in the available datasets is classical mean–variance portfolio optimization with a quadratic objective and linear constraints, e.g. $$ \begin{aligned} &\min_{w} && \sum_{i=1}^n \sum_{j=1}^n w_i w_j \sigma_{ij} \ &\text{s.t.} && \sum_{i=1}^n w_i r_i \ge R, \ & && \sum_{i=1}^n w_i = 1, \ & && l_i \le w_i \le u_i\ \forall i. \end{aligned} $$ whose description and structure are quite different from the synthetic second-order cone instances in the test set. In practice, hints or error patterns derived from such portfolio problems do not transfer to the benchmark’s second-order templates.
> > >
> > > Because we build upon publicly available datasets in which second-order problems are extremely scarce and not representative of the nonlinear test instances, we view a thorough treatment of these problems as out of scope for this work. That said, our framework is general: given a sufficiently rich set of similar nonlinear problems in the training data, we expect that the same error-analysis-and-hints pipeline could be applied to derive effective hints and substantially reduce these errors as well.

---

### Official Review · Reviewer_a5d2 · 2025-10-31

**Soundness:** 2
**Presentation:** 2
**Contribution:** 2
**Rating:** 2
**Confidence:** 4

**Summary:**

This paper proposes the OptiMind method to improve the accuracy of LLMs in modeling optimization problems.

**Strengths:**

`Hint` is a good idea.

**Weaknesses:**

- The novelty of this paper is questionable. Similar work already exists, such as LLMOPT, ORLM and SIRL, many of which utilize intermediate inference processes and mathematical modeling to guide the gradual generation of correct solution code. Furthermore, related works have investigated components like self-correction and self-debug, similar to the self-correction proposed in this paper. Therefore, the novelty of the "multi-turn inference pipeline" is questionable.
- The proposed "domain-informed error analysis framework" relies too heavily on manual annotation by domain experts, involving a large amount of manual annotation in both error summary and training data cleaning. This will significantly limit the application of OptiMind in general problem domains.
- Too many works have claimed to have cleaned the evaluation set and provided multiple different versions of ground truth for the same evaluation set, then claimed to have achieved state-of-the-art (SOTA) results on the cleaned evaluation task. I believe this is highly inappropriate. Authors should respect the contributions of related works. If problems do exist, they should provide detailed analysis and explanations of which instances in these benchmarks have what kind of problems, rather than arbitrarily proposing new ground truths, or even allowing multiple objective function values to be considered correct for a single instance. This is not a trivial matter. For example, according to my research, the OptMATH test set has 166 samples, but this paper only uses 128 of them. Where are the remaining problems? What modifications were made to these 128 samples? As far as I know, OptMATH generates problem descriptions from mathematical models, making the possibility of errors extremely low. In the results in Table 1, the reproduction results of the OptMATH dataset differ from other related works. I believe all of this is highly unrigorous.
- Unable to find the code to reproduce the article. Only the test set is included in the attachments. The reproducibility of this article is questionable.

**Questions:**

As described in weaknesses.

---

> ### Author Response · Authors · 2025-11-26
> **Official Comment by Authors (1/5)**
>
> **We thank the reviewer for their constructive feedback. Below, we address the specific concerns and questions raised, highlighting significant improvements in our experimental results and clarifying our methodological contributions. We further refer the reviewer to the general response where we discuss 1) the significant improvements in our experimental results, 2) justification our rigorous dataset cleaning, and 3) summary of our paper updates.**
>
> ### Response to Weaknesses
>
> > **Weakness 1:**  The novelty of this paper is questionable, related works have investigated components like self-correction and self-debug.
>
> We respectfully argue that our main contribution lies in the **systematic integration of human expert knowledge** to guide the entire lifecycle of the model, yielding gains that individual components cannot achieve in isolation:
>
> - **Data Cleaning:** We developed a structured pipeline to fix training data errors, which serves as the foundation of our performance. Many previous works have relied on synthetic data for training, yet few explicitly address the influence of synthetic data quality or propose solutions for improving large-scale synthetic datasets. Our ablation studies (Section 5.3, Figure 11) confirm that this specific combination of cleaning steps is necessary: training on our fully cleaned data ("full recipe") achieves a weighted average accuracy of 81.3%. In contrast, training on "Original data" yields only 67.8%, and even using "Answer-aligned data" only reaches 74%. This highlights that that rigorous cleaning and verification of synthetic data significantly boosts model accuracy and robustness compared to models trained on unrefined synthetic datasets.
>
> - **SFT Training:** We fine-tuned our model on this high-quality, cleaned data, resulting in substantial improvements over the underlying architecture. Our SFT model consistently improves 5%-21% over the GPT-OSS-20B base model across all three benchmarks in the single-turn setting. For instance, on the Mamo-Complex benchmark, our SFT increases accuracy from 64.14% (Base) to 85.52% (SFT), demonstrating that our data effectively unlocks domain-specific reasoning capabilities.
>
> - **Inference:** We utilize class-specific hints derived from expert analysis. Importantly, this mechanism is universally effective: adding hints consistently improves performance across all evaluated models, not just our own. This includes frontier models like GPT-5, which sees an accuracy boost of approximately +5% on IndustryOR (from 78.57% to 83.50%) and +4.7% on Mamo-Complex when using our hints. This confirms that our domain-informed hints provide complementary value even to the strongest existing models.
>
> Our ablation study (Figure 11) proves that this structured approach is crucial: the "full recipe" (81.3%) significantly outperforms training on "Original data" (67.8%) and even using "answer-aligned data" (74%), demonstrating that our expert-guided data cleaning pipeline is effective and essential for maximizing SFT performance.
>
> Furthermore, our updated experiments in the updated paper and General Response 1 demonstrate substantial improvements in model accuracy. Specifically, we achieve superior results compared to models of similar scale, narrowing the gap with large close-sourced models like GPT-5 while remaining more cost-effective. We believe that these improvements confirm the robustness and effectiveness of our approach.
>
>
> > **Weakness 2:** The proposed "domain-informed error analysis framework" relies too heavily on manual annotation, limiting application in general problem domains.
>
> In our data cleaning pipeline, human experts were only required to manually review and correct a small subset of problems, about 200 in total. After this initial phase, our automated process scaled up by cleaning approximately 5,000 problems without further human intervention. If new data becomes available, our pipeline will be able to clean it automatically, without any further human intervention. The domain-specific hints we employ are based on 53 canonical classes; our operations research experts have confirmed that they encompass the majority of common optimization problems. While our pipeline is capable of categorizing problems into these classes autonomously, it was essential that basic per-class knowledge be provided by human experts and incorporated into our system to enable highly-accurate results.

---

> > ### Author Response · Authors · 2025-11-26
> > **Official Comment by Authors (2/5)**
> >
> > > **Weakness 3:** Too many works have claimed to have cleaned the evaluation set.
> >
> > We appreciate the reviewer raising this concern and fully agree that repeated “cleaning” of the same benchmarks can create confusion if not done transparently. Our intention is **not** to replace or invalidate existing cleaned versions, but to provide a *fully documented* alternative that can be inspected and reused by the community.
> >
> > First, we updated the related work section to explicitly acknowledge all concurrent efforts that propose corrected versions of IndustryOR/MAMO-Complex (SIRL[1],  LogiOR [2] and Xiao et. al.[3]), and we view these as in parallel to our own work. For IndustryOR in particular, our optimization experts manually reviewed every instance where our cleaned version differs from the SIRL version by going back to the original source problems. In several cases we found that the SIRL formulation still have incorrect ground truth or ambiguous problem description, whereas our version corrects these issues; the table below presents five representative side-by-side examples. The following table lists a short explanation of the discrepancy for each such instance. For other discrepancies, the two versions reflect different but reasonable interpretations, and we do not claim that our fixes are uniquely correct.
> >
> > | Problem idx | Why SIRL version was wrong                                                                                                                                                                                                                                                                       | Error Type  |
> > |------------|---------------------------------------------------------------------------------------------------------------------------------------------------------------------------------------------------------------------------------------------------------------------------------------------------|-----------------------|
> > | 45         | Incorrect ground truth. Solution that achieves our value: Product A: 62 units (40 units at profit 10, 22 units at profit 9), Product B: 19 units (at profit 6), Product C: 0 units.                                                                                                               | Incorrect ground truth |
> > | 70    | Ambiguous description: The problem says "departs from a certain city, visits two other cities to sell merchandise", that implies visiting 3 cities instead of 4 for TSP. Also it's not clear in the table the "from city" and "to city" for the distances. We clarified both.  | Ambiguity  |
> > | 76    | The problem asks to maximize annual profit, but it doesn't specify the unit. One could return it in yuan, while the ground truth expects units of 10,000 yuan.                                                                                                                                   | Ambiguity  |
> > | 91   | Incorrect ground truth. SIRL ground truth 408.9 profit comes from a rounded LP relaxation.                                                                                                                                                                                                        | Incorrect ground truth |
> > | 98   | Incorrect ground truth. Can produce A = 15, B = 9, overtime = 81.                                                                                                                                                                                                                                 | Incorrect ground truth |
> >
> > Second, to address the concern about “arbitrarily proposing new ground truths” or “allowing multiple objective values,” we emphasize that:
> > - We **never** modify labels to favor our own model; all changes are solver-verified and applied uniformly to all methods.
> > - For each problem we change or remove, we provide an index-level log and a short reason (missing data, ambiguity, infeasibility, incorrect ground truth, etc.), together with before/after model files in the supplementary material.
> > - We allow a single objective value per instance and no longer keep multiple ground truths. For the small subset of problems where the original description was genuinely ambiguous (e.g., whether product amounts should be restricted to integers or allowed to be continuous in production planning problems), we resolve the ambiguity by explicitly clarifying the modeling choice in the problem text and then fixing one canonical objective value. This is conceptually related to the SIRL evaluation protocol, which treats variable-type mismatches leniently by flipping variables between continuous and integer when an answer mismatch occurs; however, our treatment is more conservative, as we adjust the problem specification once after our expert inspection and then use the same clarified ground truth for all methods. All such instances are explicitly tagged in our release and are handled identically for baselines and our models.

---

> > > ### Author Response · Authors · 2025-11-26
> > > **Official Comment by Authors (3/5)**
> > >
> > > Finally, to ensure that our performance claims are not tied to “our” ground truth only, we also benchmark GPT-5, o4-mini, GPT-OSS-20B, SIRL and our SFT model **on the cleaned benchmarks released by SIRL[1], LogiOR[2], and (Xiao et. al. 2025)[3]**. As shown in the table below, the trends closely match our main results: our SFT model consistently outperforms GPT-OSS-20B and SIR-Gurobi32B, and with class-specific hints and multi-turn correction it approaches GPT-5 and o4-mini even on *their* cleaned test sets. This suggests that our gains are not an artifact of using our own cleaned evaluation set, but transfer to independently cleaned benchmarks as well.
> > >
> > > *SIRL stands for Chen et al. (2025) [1], LogiOR stands for Yang et al. (2025a) [2], and Survey stands for Xiao et al. (2025a) [3].*
> > >
> > > | Model | K | Hints | IndustryOR-SIRL (1 turn) | IndustryOR-SIRL (5 turns) | IndustryOR-LogiOR (1 turn) | IndustryOR-LogiOR (5 turns) | IndustryOR-Survey (1 turn) | IndustryOR-Survey (5 turns) | MAMO-Complex-SIRL (1 turn) | MAMO-Complex-SIRL (5 turns) | MAMO-Complex-Survey (1 turn) | MAMO-Complex-Survey (5 turns) |
> > > |-----------------|---|:-----:|--------------------------:|---------------------------:|----------------------------:|-----------------------------:|----------------------------:|-----------------------------:|----------------------------:|-----------------------------:|------------------------------:|-------------------------------:|
> > > | GPT-5 | 1 | no | 69.67 | 71.00 | 68.29 | 70.12 | 66.67 | 68.25 | 72.57 | 74.05 | 65.76 | 66.21 |
> > > | | | yes | 72.00 | 73.00 | 76.82 | 78.04 | 70.23 | 69.04 | 88.17 | 89.65 | 85.58 | 86.48 |
> > > | O4-MINI | 1 | no | 61.33 | 64.67 | 66.26 | 74.79 | 68.25 | 69.84 | 57.96 | 69.95 | 58.55 | 66.67 |
> > > | | | yes | 68.67 | 70.00 | 71.54 | 74.39 | 66.67 | 63.49 | 81.77 | 86.53 | 78.82 | 86.48 |
> > > | SIRL-GUROBI32B | 1 | no | 29.25 | 29.87 | 38.90 | 39.00 | 47.85 | 47.85 | 53.05 | 53.10 | 83.06 | 82.97 |
> > > | | | yes | 30.80 | 31.20 | 35.10 | 35.60 | 46.67 | 46.67 | 65.56 | 65.61 | 77.02 | 77.20 |
> > > | GPT-OSS-20B | 1 | no | 54.30 | 62.70 | 59.87 | 65.97 | 62.85 | 67.38 | 60.39 | 77.83 | 68.10 | 78.82 |
> > > | | | yes | 57.30 | 63.30 | 66.09 | 70.00 | 62.38 | 67.38 | 73.59 | 85.27 | 77.47 | 85.85 |
> > > | OUR SFT | 1 | no | 57.30 | 63.70 | 63.71 | 67.68 | 67.61 | 68.57 | 79.80 | 85.32 | 85.22 | 85.94 |
> > > | | | yes | 59.20 | 62.60 | 67.68 | 71.64 | 69.04 | 71.42 | 84.43 | 87.68 | 87.38 | 90.09 |
> > >
> > > > **Weakness 4:** Claims regarding dataset errors are inappropriate; OptMATH claims 99.6% accuracy.
> > >
> > > We fully respect the contributions of the OptMATH authors and agree that their automatic generation pipeline substantially reduces many of labeling errors. However, our manual expert review shows that the final natural-language problem statements and ground-truth solutions are not error-free, especially when viewed from the perspective of the MILP formulation.
> > >
> > > Concretely, Appendix A.11 now includes:
> > > - Representative examples of incorrect or incomplete OptMATH instances (e.g., Problems 35 and 36, listed below), and
> > > - Error type breakdown over IndustryOR and OptMATH problems, showing how many we classify as “missing data,” “ambiguous,” “infeasible,” “incorrect ground truth,” etc.
> > >
> > > Common problems included:
> > > - 1. Missing Data: Problems referenced parameters not provided in the text .
> > > - 2. Ambiguity: Constraints were described vaguely, leading to multiple valid interpretations, so for a small number of such questions we allow multiple ground-truth objective values.
> > > - 3. Incorrect Ground-Truth.
> > >
> > > Regarding the discrepancy with the “99.6%” claim: We are somewhat baffled by the claim of 99.6% equivalence accuracy in the original OptMATH paper[4], as our manual expert review of the OptMATH benchmark revealed that many instances are wrong. Examples include:
> > >
> > > | Problem Index | Optimization Class    | Why is the original problem wrong                                                                                      | How did we fix it   |
> > > |--------------|-------------------------------------|-------------------------------------------------------------------------------------------------------------------------|-------------------------------------------------------------------------------|
> > > | 5  | Network Optimization  | Ground truth was wrong (just the variable costs are more than the initially provided ground truth) | We updated the ground truth |
> > > | 55, 67, 84, 88, 95, 131  | Set Cover  | The problem is about selecting towers to cover regions, but the description is missing data about which towers cover which regions | Generated the missing data semi-automatically, followed by manual verification |
> > > | 13, 16, 22, 23, 29, 31, 32, 36, 39, 43, 51, 54, 58, 60, 62, 63, 77, 81, 87, 97, 98, 100, 104, 105, 116, 117, 121, 127  | Job shop   | It was missing the number of machines and there was only a partial assignment of jobs to machines | Generated the missing data semi-automatically, followed by manual verification |

---

> > > > ### Author Response · Authors · 2025-11-26
> > > > **Official Comment by Authors (4/5)**
> > > >
> > > > > **Weakness 5:** Why keeping only 128 problems from OptMATH?
> > > >
> > > > We agree that dropping instances from a benchmark must be justified carefully. We start from all 166 OptMATH test problems and proceed as follows:
> > > >
> > > >
> > > > 1. We **remove 38 problems**:
> > > >    - 3 “static-line planning” instances that our experts judged to be completely underspecified (key parameters and constraints are missing to the point that the problems are both ambiguous and infeasible under any reasonable interpretation), and
> > > >    - 35 second-order conic programming problems that are highly synthetic and poorly aligned with the rest of the dataset. We delibrately omit them for the following reasons:
> > > >
> > > >     First, all second-order test problems in the benchmark are essentially copies of the same synthetic template. For example, a typical instance asks to *“determine the optimal levels of five resources ( x[0] - x[4] ) and auxiliary variables $y_i$ and $t_i$ to minimize a linear cost subject to 18 linear and second-order cone constraints”*, with every coefficient explicitly enumerated. In addition, these instances suffer from severe missing-data issues: many problem statements mention, for example, the OptMATH problems with index 142, 148, 150 and 154 mentioned “12 linear constraints” but only specify parameter values for one or two constraints, so the problems are incomplete and require substantial manual reconstruction by human labelers before they can be meaningfully used.
> > > >
> > > >     Second, when we do run our models on these nonlinear instances, we find that the errors are very concentrated: the solver typically encodes $\|y_k\|_2 \leq t_k^2 $ constraints with $t_k$ unrestricted, making the feasible region a nonconvex union of two cones. This is essentially modeling issue that we believe would be straightforward to fix if similar second-order training examples (with correctly encoded SOCs and linking equations) were available. However, **we do not find any comparable second-order problems of this type in our training data**. This means we cannot derive targeted, class-specific hints from the training corpus to address these errors, in contrast to the MILP classes where our method is designed to operate.
> > > >
> > > >     The closest problem family we see in the available training datasets is classical mean–variance portfolio optimization with a quadratic objective and linear constraints, e.g. $$ \begin{aligned} &\min_{w} && \sum_{i=1}^n \sum_{j=1}^n w_i w_j \sigma_{ij} \ &\text{s.t.} && \sum_{i=1}^n w_i r_i \ge R, \ & && \sum_{i=1}^n w_i = 1, \ & && l_i \le w_i \le u_i\ \forall i. \end{aligned} $$ whose description and structure are quite different from the synthetic second-order cone instances in the test set. In practice, hints or error patterns derived from such portfolio problems do not transfer to the benchmark’s second-order templates.
> > > >
> > > >     Because we build upon publicly available datasets in which second-order problems are extremely scarce and not representative of the nonlinear test instances, we view a thorough treatment of these problems as out of scope for this work. That said, our framework is general: given a sufficiently rich set of similar nonlinear problems in the training data, we expect that the same error-analysis-and-hints pipeline could be applied to derive effective hints and substantially reduce these errors as well.
> > > >
> > > > 2. For the **remaining 128 problems**, we keep all original indices and provide a detailed changelog indicating which problems are unchanged and which have been corrected (and how). Many of these require no modification; when we do change a problem, it is for clearly documented reasons such as missing data, inconsistent units, or incorrect ground truth, as summarized in Appendix A.11.

---

> > > > > ### Author Response · Authors · 2025-11-26
> > > > > **Official Comment by Authors (5/5)**
> > > > >
> > > > > > **Weakness 6:** Discrepancies on our Table 1 and numbers reported from paper.
> > > > >
> > > > > We thank the reviewer for pointing out that our numbers differ from some prior reports. The main reasons is that we evaluate on our cleaned version of benchmarks rather than the original version / the version cleaned by other works. We will clarify this explicitly in the caption and text and will release our evaluation scripts so that others can reproduce our numbers under exactly the same assumptions.
> > > > >
> > > > >
> > > > > > **Weakness 7:** Unable to find the code to reproduce the article.
> > > > >
> > > > > We are committed to reproducibility. We will open-source the model checkpoints, the evaluation code, and the full cleaned training datasets upon acceptance.
> > > > >
> > > > >
> > > > > ### References
> > > > > [1] Chen, Yitian, et al. "Solver-Informed RL: Grounding Large Language Models for Authentic Optimization Modeling." arXiv preprint arXiv:2505.11792 (2025).
> > > > >
> > > > > [2] Yang, Beinuo, et al. "Automated Optimization Modeling through Expert-Guided Large Language Model Reasoning." arXiv preprint arXiv:2508.14410 (2025).
> > > > >
> > > > > [3] Xiao, Ziyang, et al. "A survey of optimization modeling meets llms: Progress and future directions." arXiv preprint arXiv:2508.10047 (2025).
> > > > >
> > > > > [4] Lu, Hongliang, et al. "Optmath: A scalable bidirectional data synthesis framework for optimization modeling." arXiv preprint arXiv:2502.11102 (2025).

---

### Official Review · Reviewer_UE6w · 2025-11-01

**Soundness:** 2
**Presentation:** 4
**Contribution:** 3
**Rating:** 2
**Confidence:** 4

**Summary:**

In this paper, the authors propose a pipeline to improve the accuracy of formulating optimization problems by large language models (LLMs). The pipeline consists of (1) cleaning the training and testing data, (2) supervised fine-tuning, and (2) multi-turn inference with error-aware prompting and majority voting.

The supervised fine-tuning is standard. Hence, The main contributions are in the data cleaning and the inference, and can be summarized as follows.
* Cleaning of the three most challenging datasets (IndustryOR, MAMO Complex, and OptMATH).
* Summary of common errors made by LLMs in formulating optimization problems.
* Incorporating the error analysis above in the prompts during inference.

Overall, I appreciate the authors' efforts in having operations research (OR) experts clean the datasets and analyze the errors commonly made by LLMs. I think these efforts are of great value to the community in creating more credible datasets and benchmarks.

However, I personally think the technical depth may not be enough for a conference like ICLR. In addition, the experimental results do not strongly support some of the claims by the authors. These limitations result in my current rating.

BTW: I also think this paper might be more appropriate for the dataset and benchmark track, because the contributions in improving current datasets and benchmarks are important, but the technical novelty seems limited.

**Strengths:**

+ The authors have the three most challenging datasets cleaned by OR experts (PhDs and researchers, according to the authors). The OR experts also analyze the common errors made by LLMs for each type of problems. These efforts should be appreciated.
+ The authors submitted the cleaned test datasets in the supplementary materials.

**Weaknesses:**

- I would like to see more detailed breakdown and statistics of the errors in the three datasets of IndustryOR, MAMO Complex, and OptMATH. The authors have some vague claims that "30%-60% test instances being incorrect" `[Lines 187-188]`. However, in the OptMATH paper `[R1]`, the authors claimed that "manual analysis confirming 99.6% equivalence accuracy across all triplets". Which claim is true? Since the paper builds on the premise of high error rates of existing benchmarks, it is important to make this evidence clear.
- While cleaning datasets is important, it seems to be mostly manual. So there is limited technical novelty in cleaning datasets, analyzing errors, incorporating them into prompts, SFT, and multi-turn inference.
- The experimental results do not really support the claims by the authors.
  - The accuracy of the proposal methods is worse than GPT (`Figure 5 and Table 1`), which begs the question of how valuable the fine-tuned LLM is. Note that the comparison is done under the condition of *no* majority voting ($K=1$) for GPT models and *having* majority voting ($K=8$) for the proposed model.
  - Ablation results do not show big improvements from SFT (`Table 2 in Appendix A.1`).

Minor comments:
- In the analysis of error types in `Appendix A.10`, I think "infeasible problems" and "out-of-scope problems" (i.e., nonlinear optimization) should not be counted as errors. A problem *can* be infeasible, and the LLM should correctly formulate it and return the solution as "infeasible". Nonlinearity is also not an error in problem description.
- The authors limit the scope to mixed integer linear programs (MILP). Why is the limitation? I thought the methods could be generalized to nonlinear problems, which are more challenging and arguably more valuable.
- The author mentioned that OptMATH uses DeepSeek-V3 in `Section 4.3`, which is not true. They just used Llama.

[R1] Lu H, Xie Z, Wu Y, Ren C, Chen Y, Wen Z. "OptMATH: A Scalable Bidirectional Data Synthesis Framework for Optimization Modeling," *ICML 2025*.

**Questions:**

Please refer to the points raised in "Weaknesses".

---

> ### Author Response · Authors · 2025-11-26
> **Official Comment by Authors (1/3)**
>
> **We thank the reviewer for their constructive feedback. Below, we address the specific concerns and questions raised, highlighting significant improvements in our experimental results and clarifying our methodological contributions. We further refer the reviewer to the general response where we discuss 1) the significant improvements in our experimental results, 2) justification our rigorous dataset cleaning, and 3) summary of our paper updates.**
>
> ## Response to Weaknesses and Questions
>
> > **Weakness 1:** Lack of detailed breakdown of errors and conflicting claims on OptMATH quality.
>
> We agree that clear evidence is essential. During the rebuttal period, two additional experts at the OR-professor level reviewed the benchmark problems again to verify correctness and produce the detailed error analysis. In General Response 2, we provide illustrative examples of common error types observed in both the IndustryOR and OptMATH benchmarks, along with a summary of the distribution of these error categories. We also expanded Appendix A.11 to provide detailed statistics of error categories (missing data, ambiguities, wrong solutions, etc.) for all three original benchmarks. For comprehensive details and a step-by-step description of how we updated these benchmarks, please refer to the HTML file provided in the supplementary material.
>
> We are somewhat baffled by the claim of 99.6% equivalence accuracy in the original OptMATH paper, as our manual expert review of the OptMATH benchmark revealed that many instances are wrong. Examples include:
>
> | Problem Index | Optimization Class                 | Why is the original problem wrong                                                                                      | How did we fix it                                                              |
> |--------------|-------------------------------------|-------------------------------------------------------------------------------------------------------------------------|-------------------------------------------------------------------------------|
> | 5  | Network Optimization  | Ground truth was wrong (just the variable costs are more than the initially provided ground truth) | We updated the ground truth |
> | 55, 67, 84, 88, 95, 131            | Set Cover     | The problem is about selecting towers to cover regions, but the description is missing data about which towers cover which regions | Generated the missing data semi-automatically, followed by manual verification |
> | 13, 16, 22, 23, 29, 31, 32, 36, 39, 43, 51, 54, 58, 60, 62, 63, 77, 81, 87, 97, 98, 100, 104, 105, 116, 117, 121, 127            | Job shop                            | It was missing the number of machines and there was only a partial assignment of jobs to machines                       | Generated the missing data semi-automatically, followed by manual verification |
>
>
> > **Weakness 2:** The accuracy of the proposed methods is worse than GPT
>
> We have significantly updated our experimental results, and the new data now strongly support our claims. Our updated Figure 5 presents a fairer comparison between OptiMind and frontier models like o4-mini and GPT-5. In the single-turn setting with hints (our full inference recipe), our accuracy is within 0.6% on IndustryOR and shows improvements of +11.5% on Mamo-Complex and +2.6% on OptMATH over o4-mini. When leveraging a modest computational budget to apply 5-turn self-correction (without majority voting), our performance substantially improves upon o4-mini and becomes comparable with GPT-5, landing within 0.8% on OptMATH and surpassing it by +0.9% on IndustryOR and +4.6% on Mamo-Complex. It is crucial to note that GPT-5 is a much larger model, and even with five self-correction turns, our method remains more cost-effective.
>
>
> > **Weakness 3:** Ablation results do not show big improvements from SFT.
>
> We appreciate the reviewer highlighting this issue. We have updated our main results, which now show that our SFT model is substantially stronger, achieving a raw performance improvement of **5% to 21%** over the GPT-OSS-20B base model. Furthermore, new ablation results presented in Section 5.3 and Appendix A.2 (Figure 11, titled *Ablation on training-data configurations*) clearly demonstrate the high effectiveness of our full training data cleaning pipeline . When training with data processed using our full cleaning recipe, we achieved an average accuracy of 81.3% across the three benchmarks. This figure substantially outperforms key ablations like "Original data" data (67.8%) and "Answer-aligned data" (74%), unequivocally showing the significant improvement brought by our SFT training on the expertly cleaned data.

---

> > ### Author Response · Authors · 2025-11-26
> > **Official Comment by Authors (2/3)**
> >
> > > **Weakness 4:** Limited technical novelty in cleaning, error analysis, SFT, and multi-turn inference.
> >
> > We respectfully argue that our main contribution lies in the systematic integration of human expert knowledge to guide the entire lifecycle of the model, yielding gains that individual components cannot achieve in isolation:
> >
> > - **Data Cleaning:** We developed a structured pipeline to fix training data errors, which serves as the foundation of our performance. Many previous works have relied on synthetic data for training, yet few explicitly address the influence of synthetic data quality or propose solutions for improving large-scale synthetic datasets. Our ablation studies (Section 5.3, Figure 11) confirm that this specific combination of cleaning steps is necessary: training on our fully cleaned data ("full recipe") achieves a weighted average accuracy of 81.3%. In contrast, training on "Original data" yields only 67.8%, and even using "Answer-aligned data" only reaches 74%. This highlights that that rigorous cleaning and verification of synthetic data significantly boosts model accuracy and robustness compared to models trained on unrefined synthetic datasets.
> >
> > - **SFT Training:** We fine-tuned our model on this high-quality, cleaned data, resulting in substantial improvements over the underlying architecture. Our SFT model consistently improves 5%-21% over the GPT-OSS-20B base model across all three benchmarks in the single-turn setting. For instance, on the Mamo-Complex benchmark, our SFT increases accuracy from 64.14% (Base) to 85.52% (SFT), demonstrating that our data effectively unlocks domain-specific reasoning capabilities.
> >
> > - **Inference:** We utilize class-specific hints derived from expert analysis. Importantly, this mechanism is universally effective: adding hints consistently improves performance across all evaluated models, not just our own. This includes frontier models like GPT-5, which sees an accuracy boost of approximately +5% on IndustryOR (from 78.57% to 83.50%) and +4.7% on Mamo-Complex when using our hints. This confirms that our domain-informed hints provide complementary value even to the strongest existing models.
> >
> > Our ablation study (Figure 11) proves that this structured approach is crucial: the "full recipe" (81.3%) significantly outperforms training on "Original data" (67.8%) and even using "answer-aligned data" (74%), demonstrating that our expert-guided data cleaning pipeline is effective and essential for maximizing SFT performance.
> >
> > Furthermore, our updated experiments in the updated paper and General Response 1 demonstrate substantial improvements in model accuracy. Specifically, we achieve superior results compared to models of similar scale, narrowing the gap with large close-sourced models like GPT-5 while remaining more cost-effective. We believe that these improvements confirm the robustness and effectiveness of our approach.
> >
> >
> >
> >
> >
> > ## Response to Minor Comments
> >
> > > **Comment 1:** : Infeasible problems should not be counted as errors.
> >
> > The problems classified as infeasible were originally intended to be feasible inputs with non-empty objective function values in the public benchmarks. Our expert analysis determined that the original problem statement data led to an infeasible solution space. These were "broken" inputs, not intentionally infeasible problems. We fixed the input data to allow for a feasible solution, addressing the flaw in the original benchmark (more examples are in Appendix A.11.5).

---

> ### Author Response · Authors · 2025-11-26
> **Official Comment by Authors (3/3)**
>
> > **Comment 2:**  Why limit to MILPs but discard nonlinear (out-of-scope) problems?
>
> Our decision to focus on MILPs and to exclude the few nonlinear second-order instances from our main results is driven by both dataset quality and representativeness.
>
> First, all second-order test problems in the benchmark are essentially copies of the same synthetic template. For example, a typical instance asks to *“determine the optimal levels of five resources (x[0])–(x[4]) and auxiliary variables (y_i) and (t_i) to minimize a linear cost subject to 18 linear and second-order cone constraints”*, with every coefficient explicitly enumerated. In addition, these instances suffer from severe missing-data issues: many problem statements mention, for example, the OptMATH problems with index 142, 148, 150 and 154 mentioned “12 linear constraints” but only specify parameter values for one or two constraints, so the problems are incomplete and require substantial manual reconstruction by human labelers before they can be meaningfully used.
>
> Second, when we do run our models on these nonlinear instances, we find that the errors are very concentrated: the solver typically encodes $\|y_k\|_2 \leq t_k^2 $ constraints with $t_k$ unrestricted, making the feasible region a nonconvex union of two cones. This is essentially modeling issue that we believe would be straightforward to fix if similar second-order training examples (with correctly encoded SOCs and linking equations) were available. However, **we do not find any comparable second-order problems of this type in our training data**. This means we cannot derive targeted, class-specific hints from the training corpus to address these errors, in contrast to the MILP classes where our method is designed to operate.
>
> The closest problem family we see in the available datasets is classical mean–variance portfolio optimization with a quadratic objective and linear constraints, e.g. $$ \begin{aligned} &\min_{w} && \sum_{i=1}^n \sum_{j=1}^n w_i w_j \sigma_{ij} \ &\text{s.t.} && \sum_{i=1}^n w_i r_i \ge R, \ & && \sum_{i=1}^n w_i = 1, \ & && l_i \le w_i \le u_i\ \forall i. \end{aligned} $$ whose description and structure are quite different from the synthetic second-order cone instances in the test set. In practice, hints or error patterns derived from such portfolio problems do not transfer to the benchmark’s second-order templates.
>
> Because we build upon publicly available datasets in which second-order problems are extremely scarce and not representative of the nonlinear test instances, we view a thorough treatment of these problems as out of scope for this work. That said, our framework is general: given a sufficiently rich set of similar nonlinear problems in the training data, we expect that the same error-analysis-and-hints pipeline could be applied to derive effective hints and substantially reduce these errors as well.
>
>
>
>
>
> > **Comment 3:**  OptMATH uses Llama, not DeepSeek-V3, for generation.
>
> We greatly appreciate the reviewer for raising this point. We could not find any mention of LLaMA being used in the OptMATH paper; while we are aware that a separate work (ORLM) uses LLaMA as its architectural backbone, OptMATH appears to use Qwen2.5. We would also like to clarify that our reference to DeepSeek-V3 specifically concerns the back-translation step (which generates natural-language question descriptions from problem instances), where OptMATH explicitly states their Appendix C.1 (“Backtranslation Pipeline”): "In our reverse generation pipeline, we employ Deepseek-V3 as our foundation model and configure its temperature parameter to 0.8 to enhance the diversity of generated problems.”"

---

### Official Review · Reviewer_hBsu · 2025-11-01

**Soundness:** 2
**Presentation:** 2
**Contribution:** 2
**Rating:** 4
**Confidence:** 4

**Summary:**

This paper proposes OptiMind, a framework for improving large language models' ability to formulate mixed-integer linear programming (MILP) problems from natural language descriptions. The approach combines two main components: (1) systematic training data cleaning through class-based error analysis and automated data quality improvement, and (2) multi-turn inference with domain-informed prompting using class-specific hints and solver feedback. The authors evaluate their approach on three manually cleaned benchmarks (IndustryOR, Mamo-Complex, OptMATH), demonstrating an average improvement of 14 percentage points in formulation accuracy over base models.

**Strengths:**

- The extensive effort to clean both training and test data addresses a critical issue in the field. The authors' finding that 30-60% of benchmark instances contained errors is significant and their cleaned datasets will benefit the community.
- The class-based error analysis provides a structured approach to incorporating optimization expertise into LLM training and inference, moving beyond generic prompting strategies.
- The evaluation spans multiple challenging benchmarks with careful ablation studies, demonstrating consistent improvements across different base models.

**Weaknesses:**

- The paper primarily combines existing well-established techniques (SFT, self-consistency, multi-turn refinement), and while it achieves good results in domain-specific applications, the contribution in terms of algorithmic innovation or theoretical insights is relatively limited. This combinatorial improvement, though practical, may not constitute a significant methodological breakthrough.

- While the paper presents encouraging success cases, the systematic analysis of failure modes or situations where the method performs poorly is relatively limited. Additional discussion in this area would help readers gain a more comprehensive understanding of the method's applicable boundaries and potential limitations.

- The multi-turn inference approach requires generating multiple samples (K=8) and iterative corrections, which increases inference time and computational costs. While accuracy improvements are achieved, this computational overhead may affect the method's practical application value in time-sensitive or resource-constrained environments.

**Questions:**

- Given that the class-based error analysis relies on manual expert review, have you conducted inter-annotator agreement assessments? Understanding the consistency among different experts in identifying error patterns, and how such potential variations affect the method's reproducibility, would be valuable.

- How does the system handle problems that cannot be clearly categorized into the 50 predefined classes? For cross-category or boundary-ambiguous problems, what is the classification accuracy, and how does this uncertainty affect the final results?

- Could you provide more detailed analysis showing in what proportion of cases solver feedback actually helps improve solutions? Understanding the ratio of positive and negative impacts during the multi-turn correction process, as well as convergence characteristics, would help better understand the value of this component?

- How does your class-based prompting approach compare with other common prompt engineering techniques, such as few-shot examples or structured formulation like Five-Element Formulation?

---

> ### Author Response · Authors · 2025-11-26
> **Official Comment by Authors (1/3)**
>
> **We thank the reviewer for their constructive feedback. Below, we address the specific concerns and questions raised, highlighting significant improvements in our experimental results and clarifying our methodological contributions. We further refer the reviewer to the general response where we discuss 1) the significant improvements in our experimental results, 2) justification our rigorous dataset cleaning, and 3) summary of our paper updates.**
>
> ### Response to Weaknesses
>
> > **Weakness 1:** The contribution in terms of algorithmic innovation or theoretical insights is relatively limited.
>
> We respectfully argue that our main contribution lies in the systematic integration of human expert knowledge to guide the entire lifecycle of the model, yielding gains that individual components cannot achieve in isolation:
>
> - **Data Cleaning:** We developed a structured pipeline to fix training data errors, which serves as the foundation of our performance. Many previous works have relied on synthetic data for training, yet few explicitly address the influence of synthetic data quality or propose solutions for improving large-scale synthetic datasets. Our ablation studies (Section 5.3, Figure 11) confirm that this specific combination of cleaning steps is necessary: training on our fully cleaned data ("full recipe") achieves a weighted average accuracy of 81.3%. In contrast, training on "Original data" yields only 67.8%, and even using "Answer-aligned data" only reaches 74%. This highlights that that rigorous cleaning and verification of synthetic data significantly boosts model accuracy and robustness compared to models trained on unrefined synthetic datasets.
>
> - **SFT Training:** We fine-tuned our model on this high-quality, cleaned data, resulting in substantial improvements over the underlying architecture. Our SFT model consistently improves 5%-21% over the GPT-OSS-20B base model across all three benchmarks in the single-turn setting. For instance, on the Mamo-Complex benchmark, our SFT increases accuracy from 64.14% (Base) to 85.52% (SFT), demonstrating that our data effectively unlocks domain-specific reasoning capabilities.
>
> - **Inference:** We utilize class-specific hints derived from expert analysis. Importantly, this mechanism is universally effective: adding hints consistently improves performance across all evaluated models, not just our own. This includes frontier models like GPT-5, which sees an accuracy boost of approximately +5% on IndustryOR (from 78.57% to 83.50%) and +4.7% on Mamo-Complex when using our hints. This confirms that our domain-informed hints provide complementary value even to the strongest existing models.
>
> Our ablation study (Figure 11) proves that this structured approach is crucial: the "full recipe" (81.3%) significantly outperforms training on "Original data" (67.8%) and even using "answer-aligned data" (74%), demonstrating that our expert-guided data cleaning pipeline is effective and essential for maximizing SFT performance.
>
> Furthermore, our updated experiments in the updated paper and General Response 1 demonstrate substantial improvements in model accuracy. Specifically, we achieve superior results compared to models of similar scale, narrowing the gap with large close-sourced models like GPT-5 while remaining more cost-effective. We believe that these improvements confirm the robustness and effectiveness of our approach.
>
>
>
>
> > **Weakness 2:** The systematic analysis of failure modes is relatively limited.
>
> Thank you for the constructive feedback. We discuss some potential failure modes of our system here: first, when a given description cannot be matched to one of the 53 problem classes, then that instance cannot benefit from our hints at inference time. We note however that the fraction of instances that cannot be classified is less than 1%.
>
> While our current training data already covers a broad range of practically important classes (50+ categories, including vehicle routing, scheduling, network optimization, and others), some problem types remain underrepresented in the public datasets we rely on. These underrepresented classes can be more challenging, especially since our present effort focuses on cleaning and leveraging existing resources rather than constructing a new dataset from scratch. As a next step, we plan to gradually broaden the training distribution and enrich our class/hint library so that the model can better accommodate a wider variety of problem formulations.

---

> > ### Author Response · Authors · 2025-11-26
> > **Official Comment by Authors (2/3)**
> >
> > > **Weakness 3:** The multi-turn inference approach increases inference time and computational costs.
> >
> > Our updated result is as shown in the table below, even without test-time scaling (e.g., majority voting or multi-turn inference), our fine-tuned model already outperforms open-source LLMs of similar size (GPT-OSS-20B, Qwen3-32B) and the state-of-the-art SIRL-Gurobi32B on optimization formulation tasks, and performs on par with closed-source models such as gpt-o4-mini. Figure 9 in the updated paper shows that test-time scaling improves all models, while our relative advantage remains. In practice, users can select the amount of test-time scaling based on the available compute, and our model should retain its performance lead across different budgets.
> >
> >
> >
> > | Model | Hints | IndustryOR (1 turn) | IndustryOR (5 turns) | Mamo-Complex (1 turn) | Mamo-Complex (5 turns) | OptMATH (1 turn) | OptMATH (5 turns) |
> > | :--- | :--- | :--- | :--- | :--- | :--- | :--- | :--- |
> > | **GPT-5** | no | $78.57$ | $81.12$ | $87.86$ | $89.36$ | $90.15$ | $92.18$ |
> > | | yes | $83.50$ | $85.20$ | $92.57$ | $93.42$ | $89.21$ | $90.31$ |
> > | **o4-mini** | no | $76.19$ | $82.14$ | $75.95$ | $87.78$ | $80.31$ | $87.18$ |
> > | | yes | $82.14$ | $85.88$ | $88.09$ | $91.67$ | $84.21$ | $90.46$ |
> > | **Qwen3-32B** | no | $64.28$ | $69.39$ | $66.47$ | $77.14$ | $79.01$ | $79.49$ |
> > | | yes | $69.13$ | $74.23$ | $78.00$ | $84.76$ | $76.95$ | $80.85$ |
> > | **SIRL-Gurobi32B** | no | $51.36$ | $55.27$ | $64.00$ | $72.67$ | $70.93$ | $75.46$ |
> > | | yes | $49.38$ | $53.06$ | $75.61$ | $78.57$ | $74.06$ | $76.71$ |
> > | **GPT-OSS-20B** | no | $65.45$ | $75.80$ | $64.14$ | $88.29$ | $77.34$ | $88.05$ |
> > | | yes | $70.99$ | $78.71$ | $73.87$ | $90.61$ | $76.00$ | $87.72$ |
> > | **OUR SFT** | **no** | $70.71$ | $77.55$ | $85.52$ | $91.09$ | **$82.65$** | $88.43$ |
> > | | **yes** | **$75.61$** | **$79.48$** | **$87.42$** | **$92.42$** | $82.58$ | **$89.28$**
> >
> >
> > ### Response to Questions
> >
> > > **Q1:** Have you conducted inter-annotator agreement assessments?
> >
> > To ensuring reproducibility and high quality, each problem and hint was reviewed by at least four optimization experts (PhD level or above). During the rebuttal period, two additional experts at the OR-professor level reviewed the benchmark problems again to verify correctness and produce the detailed error analysis. We further refer the reviewer to General Response 2 for more discussions.
> >
> > > **Q2:** Could you provide more detailed analysis showing in what proportion of cases solver feedback actually helps improve solutions?
> >
> > Updated Figures 9 and 10 demonstrate that multi-turn inference consistently improves accuracy for both base and SFT models when error analysis hints are included. The gains for base models are substantial (10–20%), while SFT models see smaller but consistent improvements (3–6%). Notably, performance largely converges at Turn 3. This confirms that multi-turn inference is a valuable strategy for improving robustness and enabling test-time scaling when the inference budget permits.
> >
> > > **Q3:** How does your class-based prompting approach compare with other common prompt engineering techniques, such as few-shot examples?
> >
> > To address the reviewer’s suggestion, we ran an additional experiment with few-shot prompting. For each test problem, we retrieved the five most similar questions (by edit distance) from the training set within the same problem type and provided their cleaned solutions as exemplars. The table below reports results for the gpt-oss-20b model, comparing this few-shot setting with our succinct hint-based prompting across the three benchmarks. In all cases, our concise, targeted hints significantly outperform the few-shot examples, proving the benefit of our error-guided prompting strategy.
> >
> > | Hint Strategy | IndustryOR (1 turn) | IndustryOR (5 turns) | MAMO-Complex (1 turn) | MAMO-Complex (5 turns) | OptMATH (1 turn) | OptMATH (5 turns) |
> > |--------------------|:-------------------:|:--------------------:|:---------------------:|:----------------------:|:----------------:|:-----------------:|
> > | No hint | 65.45 | 75.8 | 63.17 | 88.33 | 77.34 | 88.05 |
> > | With hint | 70.99 | 78.71 | 73.49 | 90.63 | 76.04 | 87.72 |
> > | Few-shot examples | 67.34 | 75.40 | 66.23 | 85.85 | 77.18 | 86.79 |

---

> > > ### Author Response · Authors · 2025-11-26
> > > **Official Comment by Authors (3/3)**
> > >
> > > > **Q4:** How does the system handle problems that cannot be clearly categorized into the 50 predefined classes? For cross-category or boundary-ambiguous problems, what is the classification accuracy, and how does this uncertainty affect the final results?
> > >
> > > As our 53 predefined classes are adopted from the OptMATH seed classes, it is indeed a great question whether they cover a sufficiently wide range of problems and whether they can generalize to other benchmark datasets.
> > >
> > > In our implementation, we explicitly allow the GPT-OSS classifier to propose a new class when none of the existing 53 labels is appropriate. For example, for some second-order problems in the original OptMATH dataset that do not resemble any of the seed classes, the classifier labels them as “second-order optimization problem,” a new class that does not appear in the original OptMATH taxonomy. For such newly created classes, we currently do not provide any hint template during inference. Excluding the second-order optimization problems that we deliberately discard from our main experiments, fewer than 1% of benchmark instances fall outside the training classes, so this limitation has negligible impact on our present results.
> > >
> > > In Sec 5.3 and Table 3 in Appendix A.2 (copied below), we compare the MAMO-Complex performance with our gpt-oss based classification against several alternatives, including human expert manual classification, which we use as an approximate ground truth oracle to compare the performance.
> > >
> > > We see that our gpt-oss based automatic classification even outperforms human-expert manual classification in terms of achieving higher final accuracy performance, and hence we believe it serves as a good signal for validating the effectiveness of our classification method.
> > >
> > > Based on our discussion with human OR experts, we believe the 53 classes seem to cover a wide range of optimization problems (especially for the existing benchmarks), and hence we adopt them in this current work. However, we do acknowledge that, if there are problems that fall completely out of scope from all these 53 classes (which has less than 1% in the benchmarks), our SFT model or the hint-based analysis may not be effective. While we have not detected this to be a practical issue in our current experiments, we view it as natural future work to further extend the class coverage and improve generalization.
> > >
> > > *Note that we do not have exact ground truth class for each benchmark problem exactly because each problem may not fall exactly into one class, and hence we need to resort to human expert to label based on their best judgement.*
> > >
> > > | Classification / Hint Strategy | 1 turn | 5 turn |
> > > |--------------------------------------|:------:|:------:|
> > > | GPT-OSS classifier (ours) | 83.81 | 89.90 |
> > > | Human expert manual classification | 81.04 | 89.61 |
> > > | All hints, no classification | 78.57 | 89.23 |
> > > | Random classes | 72.57 | 87.23 |
> > > | No hints | 70.00 | 86.57 |

---

### Author Response · Authors · 2025-11-26
**General Response to Reviewers (1/3)**

We thank the reviewer for their constructive feedback. Since the initial submission, we have performed a comprehensive update to our work, resulting in two major improvements:



### **1. Significantly Improved Model Performance:**

We performed a detailed grid search for our training hyperparameters and further improved our dataset quality. As a result, our updated SFT model (based on GPT-OSS-20B) is substantially stronger than the previous version. **OptiMind** now achieves performance comparable to frontier closed-source models (like GPT-5) while clearly outperforming other open-source models of similar size. Our SFT model consistently improves **5%-21%** over the GPT-OSS-20B base model, **6%-19%** over Qwen3-32B, and **12%-21%** over the domain-specific state-of-the-art model SIRL-Gurobi32B across all three benchmarks. Combining SFT with error-analysis hints, OptiMind performs comparably to **o4-mini** in single-turn settings. With 5-turn self-correction, it outperforms iven **GPT-5**, exceeding it by **+0.9%** on IndustryOR and **+4.6%** on Mamo-Complex. See the detailed results in the table below:





| Model | Hints | IndustryOR (1 turn) | IndustryOR (5 turns) | Mamo-Complex (1 turn) | Mamo-Complex (5 turns) | OptMATH (1 turn) | OptMATH (5 turns) |
| :--- | :--- | :--- | :--- | :--- | :--- | :--- | :--- |
| **GPT-5** | no | $78.57 \pm 3.09$ | $81.12 \pm 1.79$ | $87.86 \pm 1.55$ | $89.36 \pm 1.92$ | $90.15 \pm 0.69$ | $92.18 \pm 0.78$ |
| | yes | $83.50 \pm 2.84$ | $85.20 \pm 1.25$ | $92.57 \pm 1.19$ | $93.42 \pm 0.62$ | $89.21 \pm 1.78$ | $90.31 \pm 1.04$ |
| **o4-mini** | no | $76.19 \pm 1.05$ | $82.14 \pm 2.11$ | $75.95 \pm 2.38$ | $87.78 \pm 1.15$ | $80.31 \pm 2.02$ | $87.18 \pm 0.89$ |
| | yes | $82.14 \pm 2.47$ | $85.88 \pm 1.19$ | $88.09 \pm 2.02$ | $91.67 \pm 1.52$ | $84.21 \pm 2.17$ | $90.46 \pm 1.16$ |
| **Qwen3-32B** | no | $64.28 \pm 3.00$ | $69.39 \pm 3.00$ | $66.47 \pm 1.64$ | $77.14 \pm 3.18$ | $79.01 \pm 2.41$ | $79.49 \pm 1.84$ |
| | yes | $69.13 \pm 2.00$ | $74.23 \pm 3.85$ | $78.00 \pm 1.73$ | $84.76 \pm 2.12$ | $76.95 \pm 1.35$ | $80.85 \pm 1.35$ |
| **SIRL-Gurobi32B** | no | $51.36 \pm 1.39$ | $55.27 \pm 1.63$ | $64.00 \pm 1.41$ | $72.67 \pm 1.37$ | $70.93 \pm 1.94$ | $75.46 \pm 1.62$ |
| | yes | $49.38 \pm 3.02$ | $53.06 \pm 3.61$ | $75.61 \pm 2.05$ | $78.57 \pm 0.95$ | $74.06 \pm 1.39$ | $76.71 \pm 2.30$ |
| **GPT-OSS-20B** | no | $65.45 \pm 4.79$ | $75.80 \pm 3.26$ | $64.14 \pm 4.24$ | $88.29 \pm 1.26$ | $77.34 \pm 3.49$ | $88.05 \pm 1.33$ |
| | yes | $70.99 \pm 1.84$ | $78.71 \pm 2.07$ | $73.87 \pm 2.43$ | $90.61 \pm 1.36$ | $76.00 \pm 2.24$ | $87.72 \pm 1.95$ |
| **OUR SFT** | **no** | $70.71 \pm 3.15$ | $77.55 \pm 1.18$ | $85.52 \pm 2.56$ | $91.09 \pm 1.34$ | **$82.65 \pm 1.83$** | $88.43 \pm 1.46$ |
| | **yes** | **$75.61 \pm 2.65$** | **$79.48 \pm 2.28$** | **$87.42 \pm 1.05$** | **$92.42 \pm 0.96$** | $82.58 \pm 1.60$ | **$89.28 \pm 0.86$** |


Beyond the main results, we also conduct a broad set of ablation studies to disentangle the effects of data cleaning and hint (See Section 5.3 for details). We compare training on the original versus our cleaned datasets, and evaluate across varing the hint strategies. Across all three benchmarks, these ablations consistently show that (i) trainin on our cleaned data provide a substantial boost over the raw data, and (ii) class-specific hints yield robust gains.
As an additional check, we also tried a simple few-shot prompting variant: for each test problem, we retrieved the five most similar training questions (by edit distance) within the same problem type and provided their cleaned solutions as exemplars. This few-shot approach consistently underperforms our succinct, hint-based prompts, suggesting that class-specific hints provide more effective guidance than directly reusing solved examples.

| Hint Strategy | IndustryOR (1 turn) | IndustryOR (5 turns) | MAMO-Complex (1 turn) | MAMO-Complex (5 turns) | OptMATH (1 turn) | OptMATH (5 turns) |
|--------------------|:-------------------:|:--------------------:|:---------------------:|:----------------------:|:----------------:|:-----------------:|
| No hint | 65.45 | 75.8 | 63.17 | 88.33 | 77.34 | 88.05 |
| With hint | 70.99 | 78.71 | 73.49 | 90.63 | 76.04 | 87.72 |
| Few-shot examples | 67.34 | 75.40 | 66.23 | 85.85 | 77.18 | 86.79 |

---

> ### Author Response · Authors · 2025-11-26
> **General Response to Reviewers (2/3)**
>
> ### **2. Rigorous Data Correctness:**
>
>
>
> We emphasize that our cleaned benchmarks have been manually verified by experts. While some previous works claimed correctness, we found significant residual errors (30-50%). We are now including a comprehensive list of corrections we made on each dataset and the reason for each change); see the updated supplementary material industryOR_original_vs_ours.html and optmath_original_vs_ours.html.
>
>
>
> As requested by several reviewers, we also provided a comprehensive breakdown of benchmark errors and their corresponding statistics. Below, the first table presents representative examples of errors found in IndustryOR and OptMATH, while the second table summarizes the frequency and distribution of these error types across both benchmarks.
>
>
> **IndustryOR Incorrect Problems & Fixes Examples**
>
> | Problem Index | Why is the original problem wrong                                   | How did we fix it                                              |
> |--------------|----------------------------------------------------------------------|-----------------------------------------------------------------|
> |5 | Incorrect ground truth: One could never have 1600 grams of fibres with at most 600 g of total food intake.  | We updated the ground truth |
> | 24           | The problem did not state the objective function                     | We added an objective function (minimize pollution)            |
> | 46           | The original instance was infeasible but had a ground truth of 472.3 | We updated the data to make it feasible and updated the ground truth accordingly |
>
>
> **OptMATH Incorrect Problems & Fixes Examples**
>
> | Problem Index | Optimization Class                 | Why is the original problem wrong                                                                                      | How did we fix it                                                              |
> |--------------|-------------------------------------|-------------------------------------------------------------------------------------------------------------------------|-------------------------------------------------------------------------------|
> | 5  | Network Optimization  | Ground truth was wrong (just the variable costs are more than the initially provided ground truth) | We updated the ground truth |
> | 55, 67, 84, 88, 95, 131            | Set Cover     | The problem is about selecting towers to cover regions, but the description is missing data about which towers cover which regions | Generated the missing data semi-automatically, followed by manual verification |
> | 13, 16, 22, 23, 29, 31, 32, 36, 39, 43, 51, 54, 58, 60, 62, 63, 77, 81, 87, 97, 98, 100, 104, 105, 116, 117, 121, 127            | Job shop                            | It was missing the number of machines and there was only a partial assignment of jobs to machines                       | Generated the missing data semi-automatically, followed by manual verification |
>
>
> **Error Breakdowns (High-Level)**
>
> *(Note that some instances belong to multiple error classes.)*
>
> | Benchmark  | # (%) Incorrect Problems | # (%) Missing Data | # (%) Ambiguous Problems | # (%) Infeasible | # (%) Incorrect Ground Truth | # (%) Underspecified | # (%) Specific Solver | # (%) Non-linear Problem | # (%) Inconsistent | # (%) Integral/ Fractional |
> |-----------|---------------------------|--------------------|--------------------------|------------------|------------------------------|----------------------|------------------------|--------------------------|--------------------|----------------------------|
> | IndustryOR | 50 (50%)                 | 0 (0%)             | 11 (11%)                 | 2 (2%)           | 25 (25%)                     | 5 (5%)               | 5 (5%)                 | 1 (1%)                   | 2 (2%)             | 4 (4%)                     |
> | OptMATH   | 68 (41%)                 | 48 (29%)           | 11 (7%)                  | 1 (0.6%)         | 10 (6%)                      | 1 (0.6%) | 0 (0%) | 0 (0%) | 0 (0%) | 0 (0%) |
>
>
>
> We want to point out to the reviewers that cleaning these benchmarks is nontrivial given the high initial error rate in the original data sources. We do our best to ensure rigor by applying a structured and repeatable cleaning process included multiple rounds of expert review: 4 expert labelers before the rebuttal, followed by 2 expert labelers during the rebuttal period, for a total of 4 cleaning passes. We recognize the difficulty and frustration researchers face when relying on flawed benchmark data, and our goal is to provide a reliable and transparent resource for the community. All corrections are documented in the supplementary material, and we plan to release the full cleaning logs on GitHub for broader visibility.

---

> > ### Author Response · Authors · 2025-11-26
> > **General Response to Reviewers (3/3)**
> >
> > ### **3. Summary of Paper Updates**
> >
> > The table below summarize our updates to the paper:
> >
> >
> > | Section | Summary of Update |
> > | :--- | :--- |
> > | **Abstract & Sec 1** | Updated performance claims and explicitly highlighted our data cleaning pipeline, training and error-analysis framework as our core contributions. |
> > | **Sec 3 (Related Work)** | Expanded to include recent concurrent works (SIRL, LogiOR, and Xiao et. al. (2025)) and their data cleaning efforts. |
> > | **Sec 4.2** | Update the "Training Data Cleaning" description for our refined training set cleaning process. |
> > | **Sec 4.4** | Update the description of multi-turn correction and majority voting as test-time-scaling techniques. |
> > | **Sec 5.1 (Datasets and Baselines)** | Update the accuracy metric and added *externally* cleaned benchmarks (from SIRL, LogiOR, and Xiao et al.) for completeness. |
> > | **Sec 5.2 (Main Results)** | **Major Addition:** Updated evaluation results and interpretations and showed that OptiMind's improvement is significant. Updated Fig. 5, 6, 7, 8, and 9. |
> > | **Sec 5.3 (Ablation Studies)** | Added a comprehensive "Ablation on training data" to empirically prove that the full data cleaning recipe is essential for maximize performance gains. Added “Ablation on classification” to justify the use of the GPT-OSS classifier. |
> > | **Appendix A.1** | Updated details of evaluation results, including the re-evaluation based on previous papers’ cleaned benchmarks (Tab. 1, Tab. 2 and Fig. 10). |
> > | **Appendix A.2** | Updated details of ablation studies (Tab. 3, and Fig. 11). |
> > | **Appendix A.8** | Updated descriptions for hyperparameter grid search. |
> > | **Appendix A.8** | Updated per-problem-class breakdown of the effect of hints (Fig. 12). |
> > | **Appendix A.11** | Included statistics of error types in each benchmark and OptMATH examples |
> > | **Appendix A.11.6** | Update reasons for discarding nonlinear problems |
> > | **Supplementary Material** | Add description of supplementary html files (list of corrections for benchmarks) |
> >
> >
> > We also include expanded description of our benchmark cleaning efforts in the supplementary material: In the **HTML files** of our supplementary material, you will find a table with columns *[Problem Index, Original Problem, Original Answer, Updated Problem, Updated Answer, How did we fix it]*, which clearly illustrates the structure and content of our dataset corrections.

---

### Author Response · Authors · 2025-12-02
**Message to Area Chair**

Dear Area Chair,

Thank you for considering our paper and rebuttal. We have uploaded a revised version of our paper to OpenReview, which incorporates a comprehensive update to our work, showing **significantly stronger results** and **detailed evidence** validating our data cleaning innovation and quality. We respectfully request you consider the following three key points during your decision-making process:

## 1. Significant Improvements in Model Performance


Since the initial submission, we have updated our SFT model using a refined hyperparameter search and improved dataset quality. As detailed in our General Response 1 and Section 5.2 of our paper, the results are substantial:



- **SOTA among Open Source:** Our model now outperforms open-source baselines of similar size (gpt-oss-20b, Qwen3-32B) by 6%-19% and the domain-specific state-of-the-art (SIRL-Gurobi32B) by 12%-21%.

- **Competitive with Frontier Models:** In single-turn settings, our 20B model performs comparably to GPT-4o-mini. With 5-turn self-correction, it matches or exceeds GPT-5, outperforming it by +4.6% on the challenging Mamo-Complex benchmark.



## 2. Clarification on Data Cleaning and Reviewers’ Factual Errors

A primary concern raised by Reviewers UE6w and a5d2 was the validity of our data cleaning, with some citing the original OptMATH paper’s claim of "99.6% accuracy" to suggest our cleaning was unnecessary or redaundant.

**We respectfully highlight a significant Factual Error in these reviews regarding the quality of existing benchmarks.** Despite prior claims, our re-examination by optimization experts (PhD and Professor level) revealed that **30-50% of instances in IndustryOR and OptMATH contained critical errors**, including missing parameters, ambiguous descriptions, and incorrect ground truth.

- **Evidence of OptMATH Flaws:** Contrary to the "99.6% accuracy" claim cited by reviewers, we found severe issues in OptMATH. Our General Response 2 provides 35 concrete examples of the benchmark errors.

- **Transparency:** We have provided HTML comparison files in the supplementary material  that list every single correction side-by-side with the original, including the rationale for each fix.

- **Verification:** Our cleaning process involved 4 passes of expert review. To ensure our results are not artifacts of our specific cleaning, we also evaluated our model on benchmarks cleaned by concurrent works (SIRL, LogiOR and Xiao et. al. 2025), where our model maintained its superior performance (Table 2 in Appendix A.1).

## 3. Justification of Contribution as a Systematic Pipeline

Some reviewers characterized our work as a combination of existing techniques. We argue in General Response 3 that our contribution is the **systematic pipeline** for instilling domain expertise, which yields gains that isolated components cannot achieve.

Our **new ablation studies** (Figure 11 in our paper) demonstrate that this specific combination is essential. Training on our "Full Recipe" cleaned data achieves 81.3% accuracy, whereas training on "Original Data" yields only 67.8% and "Answer-aligned data" only 74%. These results confirm that careful, domain-informed training data cleaning is essential for performance gains, and this aspect, to our knowledge, has largely been neglected in prior work.

We believe the combination of our corrected benchmarks, the 20%+ performance boost over base models, and our data cleaning and inference pipeline provides a substantial contribution to the community, and we are committed to fully open source our model, code, and data upon paper acceptance.

Thank you for your time and consideration.

---

### Meta-Review · Area_Chair_RWk5 · 2026-01-05

**Summary:**

Reviewing all the reviews and author responses, I have to recommend rejection despite the authors' substantial efforts during rebuttal.

The core issue is that this work, while showing improved empirical results after rebuttal, remains primarily an engineering effort combining existing techniques rather than offering methodological innovation. All four reviewers raised concerns about limited novelty, with the consensus being that SFT on cleaned data plus class-based prompting and multi-turn refinement are well-established practices. The authors argue their contribution lies in systematic integration, but the ablations don't convincingly show that their specific pipeline is uniquely effective compared to alternative combinations of these standard components.

More problematic is the data cleaning controversy. Multiple reviewers questioned the necessity and rigor of the benchmark corrections, especially given OptMATH's claimed 99.6% accuracy. While the authors provided detailed error lists during rebuttal, the fact that concurrent works (SIRL, LogiOR) produced different cleaned versions of the same benchmarks creates confusion rather than clarity. The community would benefit more from a careful reconciliation of these different cleaning efforts rather than yet another variant. The authors' decision to exclude 38 of 166 OptMATH problems, including all nonlinear instances, further limits the scope and makes direct comparison with prior work difficult.

The scalability concern also remains unaddressed. The pipeline required extensive expert involvement for the initial 200 problem review and hint generation. While the authors claim this then scales automatically, the 53 problem classes are essentially fixed and cover less than 1% of out-of-scope problems. Extending to new domains would require similar expert effort, limiting practical applicability.
The improved results are encouraging, but not sufficient to overcome these fundamental limitations for a venue like ICLR that values methodological contributions.

**Reviewer Concerns:**

The rebuttal successfully addressed several empirical concerns but left fundamental methodological issues unresolved.

Addressed:

Reviewer UE6w's concerns about experimental performance were largely resolved. The updated results show the SFT model now substantially outperforms baselines (5-21% gains over GPT-OSS-20B) and approaches frontier models with multi-turn inference, directly addressing the initial observation that the method underperformed GPT.

Reviewer UE6w's request for detailed error breakdowns was met comprehensively. The authors provided extensive error statistics, concrete examples from both IndustryOR and OptMATH, and HTML comparison files documenting every correction. This transparency is valuable regardless of other concerns.

Reviewer hBsu's question about computational overhead was addressed by demonstrating strong single-turn performance without requiring expensive multi-turn inference.

The few-shot comparison requested by Reviewer hBsu was conducted, showing class-based hints outperform retrieval-based prompting.

Partially Addressed:

The scalability concern (Reviewers a5d2 and 3yxj) received mixed treatment. The authors claim only 200 problems required manual review with subsequent automation, but the 53 problem classes remain fixed and required expert knowledge to establish. Extending to genuinely new domains would still require similar expert effort.

The benchmark cleaning controversy (Reviewer a5d2) was addressed with extensive documentation, but the existence of multiple independently cleaned versions (SIRL, LogiOR, Survey) actually reinforces concerns about reproducibility and standardization rather than resolving them.

Still Outstanding:

All four reviewers identified limited novelty as a core weakness. The authors' response emphasizes systematic integration and strong empirical results, but does not demonstrate that their specific pipeline represents a methodological advance over alternative combinations of standard techniques (cleaned data, SFT, prompting, multi-turn refinement). The ablations show their full recipe works, but not that it uniquely works or offers generalizable insights.

The narrow scope limitation (Reviewer 3yxj) remains. The exclusion of all 35 nonlinear problems from OptMATH due to data quality and lack of training coverage confirms rather than addresses this constraint.

**Reviewer Scores:**

Reviewer hBsu (Original: 4) - Likely would increase to 5 or remain at 4 with more positive tone. The substantial empirical improvements (5-21% gains over base model, competitive with GPT-5) directly address their concern about results. All four specific questions were answered with additional experiments. However, the core concern about limited algorithmic innovation versus systematic engineering remains unresolved, preventing a higher score.

Reviewer UE6w (Original: 2) - Likely would increase to 3 or possibly 4. The comprehensive error breakdown with statistics and examples thoroughly addresses their primary evidence-based concern about dataset quality. The dramatically improved experimental results (now outperforming similar-sized models and approaching frontier models) resolve their empirical objections. However, their fundamental concern about technical depth for ICLR and the observation that this might be better suited for a dataset track remains valid and would prevent acceptance-level scoring.

Reviewer a5d2 (Original: 2) - Likely would remain at 2 or marginally increase to 3. While the detailed error documentation addresses the evidence request, this reviewer's core concern about creating yet another competing cleaned benchmark version was actually reinforced rather than resolved by showing multiple independent cleaning efforts (SIRL, LogiOR, Survey). The novelty concerns relative to concurrent work remain unaddressed. The reproducibility issue shows commitment but doesn't change the current assessment.

Reviewer 3yxj (Original: 4) - Likely would increase to 5 or remain at 4. The improved empirical results validate the comprehensive methodology they acknowledged as a strength. However, their concerns about limited conceptual novelty, manual effort requirements, and narrow MILP scope all remain fundamentally unaddressed. The authors' response on nonlinear problems actually confirms rather than mitigates the scope limitation.

---

### Decision · Program_Chairs · 2026-01-26

Reject